# Early life experiences selectively mature learning and memory abilities

Benjamin Bessières [1,2], Alessio Travaglia[1,2], Todd M. Mowery[1], Xinying Zhang[1] & Cristina M. Alberini [1*]

The mechanisms underlying the maturation of learning and memory abilities are poorly understood. Here we show that episodic learning produces unique biological changes in the hippocampus of infant rats and mice compared to juveniles and adults. These changes include persistent neuronal activation, BDNF-dependent increase in the excitatory synapse markers synaptophysin and PSD-95, and significant maturation of AMPA receptor synaptic responses. Inhibition of PSD-95 induction following learning impairs both AMPA receptor response maturation and infantile memory, indicating that the synapse formation/maturation is necessary for creating infantile memories. Conversely, capturing the learning-induced changes by presenting a subsequent learning experience or by chemogenetic activation of the neural ensembles tagged by learning matures memory functional competence. This memory competence is selective for the type of experience encountered, as it transfers within similar hippocampus-dependent learning domains but not to other hippocampus-dependent types of learning. Thus, experiences in early life produce selective maturation of memory abilities.

[1] Center for Neural Science, New York University, New York, NY 10003, USA. [2]These authors contributed equally: Benjamin Bessières, Alessio Travaglia. *email: ca60@nyu.edu

Early-life episodic experiences are rapidly forgotten[1–10]; nonetheless, they profoundly affect the brain's functions and physiology throughout life[11–20]. In agreement with this literature, recent studies in both rats and mice have provided evidence that while infantile memories are not accessible for long-term expression they are not lost, but instead are stored long term in latent forms. In fact, these memories can be reinstated later in life by behavioral reminders or artificial reactivation of networks that were active during learning[21–23].

In recent studies using inhibitory avoidance (IA) or novel object location (nOL) learning in infant rats at postnatal day 17 (PN17, infantile learning), we showed that learning requires the dorsal hippocampus (dHC), where it induces a brain-derived neurotrophic factor (BDNF)- and mGluR5-dependent switch in the ratio of N-methyl-D-aspartic acid receptor (NMDAR) subunits GluN2A/GluN2B. These changes are necessary for the formation of long-lasting, latent memories[22]. We also found that the hippocampus at PN17, compared with PN24 (juvenile) and PN80 (adult age), has significantly higher basal levels of cellular activation and plasticity markers[24]. These proteins include the immediate early genes (IEGs) c-Fos, Zif268, and activity-regulated cytoskeleton-associated protein (Arc/Arg3.1), which in the adult hippocampus are present at low concentration but are rapidly and transiently increased in response to learning[25–27]. The activity-dependent inductions of these proteins, which in the adult brain peak 1–2 h after training and return to baseline a few hours later, are indispensable for long-term plasticity and memory formation[25–27]. The rat hippocampus at PN17 also has significantly higher levels of phosphorylation of the α-amino-3-hydroxy-5-methyl-4-isox-azoleproprionic acid receptor (AMPAR) subunit GluA1 at Ser-831 and Ser-845[24]. Activity-dependent phosphorylation of these Ser residues is important for regulating the delivery and stabilization of AMPARs at synapses and single-channel conductance, and thus for long-term plasticity[28–31]. We suggested that the higher levels of activity/plasticity proteins in baseline conditions at PN17 compared with those of more mature and especially adult ages are not the result of default developmental processes, but reflect learning-induced activations[24], in agreement with the observations that infants display high rates of learning[3,9].

Given the facts that (i) the hippocampus responds to new learning using differential regulations within a limited period in infancy, when memory abilities are developing and maturing, and (ii) these differential regulations include mechanisms typically employed by sensory systems during their critical period of functional development, we proposed that hippocampus-dependent learning, such as sensory functions, undergoes developmental critical periods[22,32]. According to this view, during the critical period, the abilities to learn and memorize like adults should be acquired.

However, little is known about the biological mechanisms induced by learning and required for the maturation of hippocampus-dependent learning and memory abilities and important questions remain to be addressed: what types of biological mechanisms are recruited for the maturation of memory abilities? Does the experience-dependent maturation develop the functions of the hippocampal system as a whole, or does it mature selective abilities, reflective of the specific learning experiences encountered? In other words, are learning and memory abilities, which are central to cognition and cognitive development, the result of selective maturation processes built on specific learning experiences? If this is the case, the implications are substantial: it would suggest that an individual's experiential history greatly shapes the subject's cognitive abilities. Addressing these problems therefore has vast implications for mental health and diseases. In this study, we tackled these questions using contextual, aversive, and spatial, nonaversive learning in infant rats or mice at PN17, artificial trace reactivation, biochemical, molecular, and electrophysiological analyses. We show that episodic learning in infants produces unique changes in neural activation and synapse formation/maturation in the hippocampus compared with juveniles and adults, including lasting IEGs activation, upregulation of the excitatory synapse markers synaptophysin and PSD-95, and maturation of AMPA receptor synaptic responses. Repetition or artificial reactivation of the initial learning experience recruits this biological development to functionally mature memory competences in domain-selective manners. Thus, experiences in early life develop selective biological and functional maturation of memory abilities.

## Results

**Infantile learning induces long-lasting neuronal activation**. To determine whether new learning in infancy matures fundamental mechanisms of neuronal functions, here we investigated the effect of episodic learning on key constituents of neuronal activation. First, we assess whether learning further changes the already high level of hippocampal cellular activation[24] by employing western blot analyses to quantify the relative levels of Zif268, Arc, and c-Fos at various time points after IA training at PN17. We found that rats euthanized at 30 min, 9 h, 24 h, 48 h, or 7 days after IA training, compared with untrained controls (i.e., rats that remained in the home cage and were euthanized at matched time points, referred to hereafter as naive rats) exhibited an unusual, gradual, lasting induction of the IEGs Arc and Zif268, which peaked at 48 h after training (Fig. 1a). Training at PN17 also increased (albeit to a lesser extent) the levels of c-Fos, which peaked at 24 h after training and returned to baseline a day later. No significant differences in the levels of IEGs were found 7 days after PN17 learning between naive and trained rats (Fig. 1a).

To control for changes that may have been induced by nonassociative experience, we used two additional control groups: (i) rats exposed to an immediate footshock without IA-context exposure (shock only) and (ii) rats exposed to the IA context without footshock (context only). Both groups were euthanized 24 h after training, a time point at which all IEGs tested were significantly induced. We observed no changes in any of the IEGs in either control group relative to naive controls (shock only, Supplementary Fig. 1; context only, Supplementary Fig. 2), leading us to conclude that the lasting increase in IEG expression after training reflects associative learning.

To determine whether these slow and lasting IEG inductions are specific to early development, limited to the critical period of infantile amnesia, we investigated the same kinetics in rats at PN24, an age at which the animals have exited the infantile amnesia period and are able to express strong and long-lasting associative memory, similar to adult rats. PN24-trained rats exhibited significant rapid and transient induction of all IEGs, like those of adult rats, with a significant peak at 30 min after training that decayed rapidly thereafter (Fig. 1a). We concluded that the rat hippocampus at PN17 responds with distinct kinetics of IEG regulation following learning.

**Synapse formation/maturation with infant learning and memory**. The slow and lasting profile of IEG induction following training at PN17 parallels that of the BDNF receptor TrkB phosphorylation and of NMDAR subunits GluN2A and GluN2B[22], suggesting that learning may result in developmental maturation and perhaps formation of new synapses. Hence, we set out to test this hypothesis and focused on excitatory synapses. As a proxy for synapse formation and maturation, we measured the levels of postsynaptic density 95 (PSD-95), a scaffolding

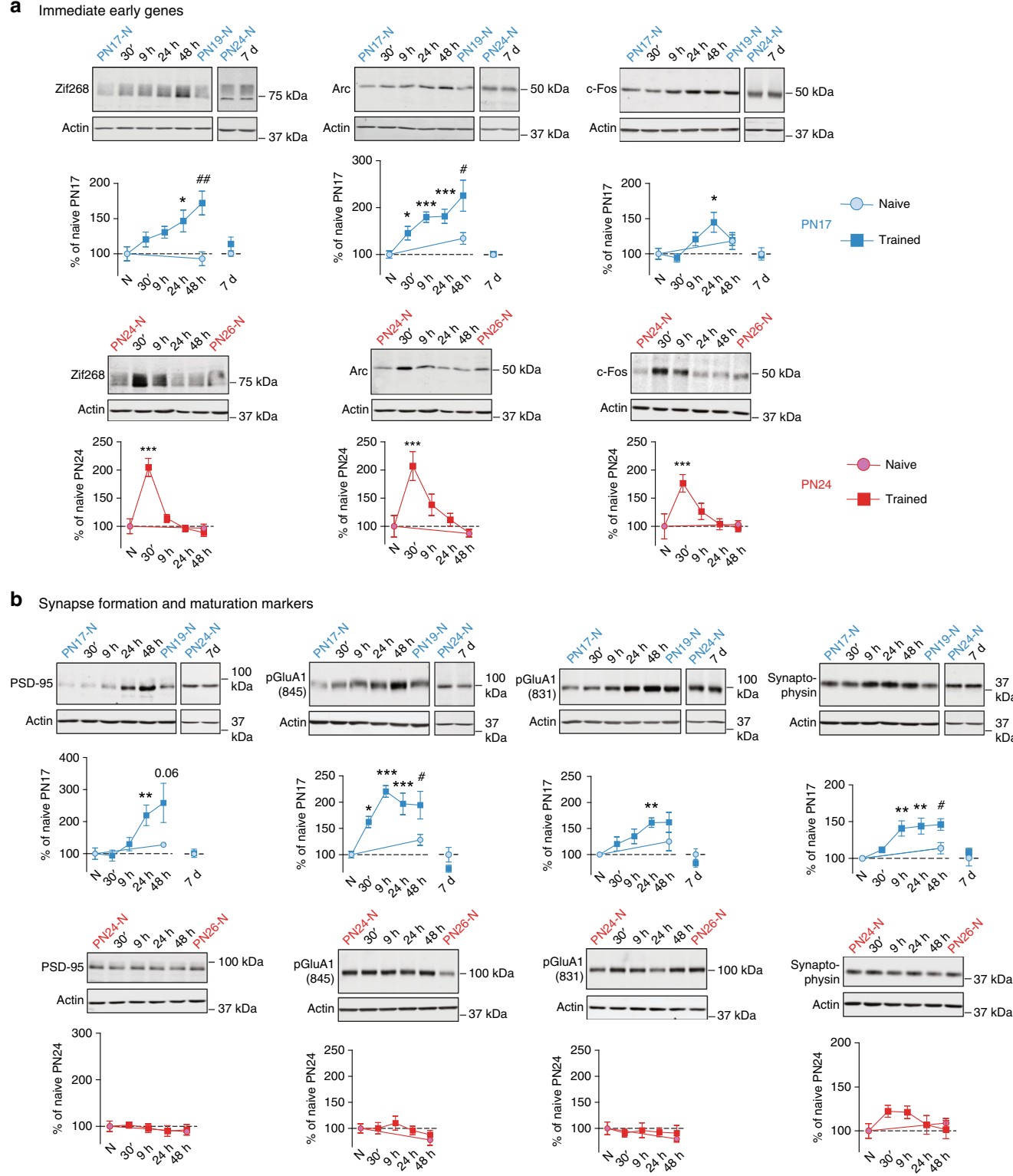

**Fig. 1 Infantile learning leads to long-lasting increases in markers of neuronal activity and synapse formation/maturation. a**, **b** Examples and densitometric western blot analyses of dHC whole-protein extracts obtained from rats trained in IA at PN17 (blue) or PN24 (red) and euthanized 30 min, 9 h, 24 h, 48 h, or 7 days after training (n = 4–10 rats per group). To account for developmental differences, four groups of naive rats were used: PN17 (n = 6–8) and PN19 (n = 6) or PN24 (n = 4–8) and PN26 (n = 6–7). **a** Immediate early genes: Zif268, Arc, and c-Fos. **b** Synapse formation/maturation markers: PSD-95, pGluA1(845), pGluA1(831), and synaptophysin. Actin was used as a loading control. Data are expressed as mean percentage ± s.e.m. of the value in the corresponding naive group (N). *P < 0.05, **P < 0.01, ***P < 0.001: significance compared with PN17 or PN24 naive rats (one-way ANOVA followed by Dunnett's multiple comparisons test); #P < 0.05, ##P < 0.01: significance comparing PN19 naive rats to 48 -h trained groups (two-tailed unpaired Student's t test). The hippocampal extracts collected 7 days after training at PN17 were analyzed separately from the other time points, and the levels of expression of the different markers were normalized on those measured in naive rats euthanized at PN24 to account for developmental differences (two-tailed unpaired Student's t test). For detailed statistical information, see Supplementary Table 1.

protein that plays critical roles in formation and maturation of new excitatory synapses by interacting with, stabilizing and trafficking NMDARs and AMPARs to the postsynaptic membrane[33,34]. We also measured the expression levels of the predominant AMPAR subunits, GluA1 and GluA2, as well as phosphorylation of GluA1 at Ser-831 and Ser-845. Finally, as a presynaptic marker of synapse formation and maturation, we assessed changes in synaptophysin, a synaptic vesicle protein critical for activity-dependent synapse formation[35,36].

IA training at PN17 significantly increased PSD-95 levels, which peaked 24 h after training and remained significantly elevated at 48 h (Fig. 1b), but did not change the overall levels of GluA1 or GluA2 (Supplementary Fig. 3). However, pGluA1 (845) was significantly induced 30 min after training and for up to 48 h afterward (Fig. 1b). pGluA1(831) was also induced after training, albeit more gradually, and was significantly elevated relative to naive rats 24 h after training. Training also significantly increased synaptophysin levels starting 9 h after training; this upregulation persisted up to 48 h after training (Fig. 1b). All changes returned to control levels by 7 days after learning (Fig. 1b). By contrast, no change in the levels of PSD-95, pGluA1(845), pGluA1(831), or synaptophysin was found following training at PN24 (Fig. 1b).

The slow and lasting increases in the levels of pGluA1(845) and pGluA1(831), IEGs, synaptophysin, and PSD-95 were consistent with similar kinetics observed previously in GluN2A and GluN2B[22], suggesting that a slow synapse formation and maturation was differentially taking place in response to learning at PN17 compared with learning at PN24.

BDNF is instrumental in synapse maturation, as well as critical periods[32,37,38] and is required in the dHC for infantile memory formation[22]. Hence, we tested whether learning-induced synapse formation and/or maturation changes require BDNF. Bilateral injection of a function-blocking anti-BDNF antibody into the dHC 30 min before training significantly disrupted the increases in both synaptophysin and PSD-95 at 24 h after training, in comparison with control IgG (Fig. 2a). By contrast, anti-BDNF antibody had no significant effect on the training-induced increase in pGluA1(845) and pGluA1(831) (Fig. 2a), indicating that BDNF is necessary for the learning-dependent increase in levels of synaptic structural proteins, but not AMPA receptor activation.

Next, we investigated whether learning-induced increase in PSD-95 is essential for infantile memory formation. The increase in PSD-95 was abolished by bilateral hippocampal injections of a PSD-95 antisense oligodeoxynucleotide (AS-ODN) (Fig. 2b) as

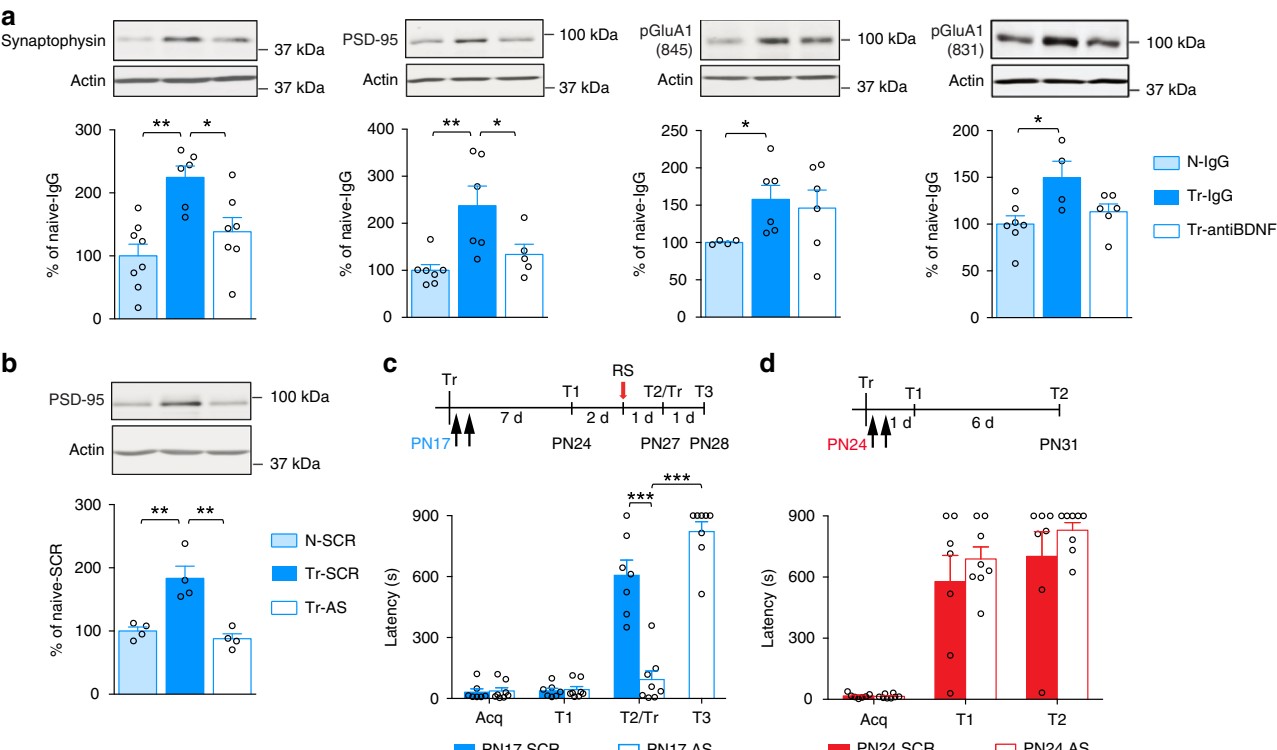

**Fig. 2 Learning-induced de novo PSD-95 synthesis is required for infantile memory formation. a** Examples and densitometric western blot analyses of synaptophysin, PSD-95, pGluA1(845), and pGluA1(831) carried out with dHC whole-protein extracts obtained from PN17 naive (N) and trained (Tr) rats, which received a bilateral hippocampal injection of IgG (N-IgG: $n = 4$–8 rats; Tr-IgG: $n = 4$–6 rats) or anti-BDNF (Tr-anti-BDNF: $n = 5$–7 rats) 30 min before training. Rats were euthanized 24 h after training. Data are expressed as mean percentage ± s.e.m. of the value in naive rats injected with IgG. *$P < 0.05$, **$P < 0.01$ (one-way ANOVA followed by Tukey's multiple comparisons test). **b** Examples and PSD-95 densitometric western blot analyses of dHC extracts obtained from naive (N) and trained (Tr) rats, which received two hippocampal injections of scrambled (SCR) ($n = 4$ rats per group) or PSD-95-antisense ODN (AS) ($n = 4$ rats), the first immediately after training at PN17 and the second 6 h later. Rats were euthanized 24 h after training. Data are expressed as mean percentage ± s.e.m. of the value in naive rats injected with SCR. **$P < 0.01$ (one-way ANOVA followed by Tukey's multiple comparison test). **c**, **d** Mean latency ± s.e.m. (in seconds, s) of rats injected twice (black arrows) in the dHC with either SCR ($n = 7$ rats) or AS ($n = 8$ rats) against PSD-95, immediately after and 6 h after training at PN17 (**c**), or at PN24 (**d**). Acq: latency to enter the dark side of the chamber at training (Tr). Rats trained at PN17 were tested 7 days later (T1), and 2 days later received a reminder shock (RS), followed by another memory retention test 1 day later (T2). At T2, upon entering the shock compartment, rats were trained again (Tr) and tested 1 day later (T3). Rats trained at PN24 were tested 1 day after training (T1) and again 6 days later (T2). ***$P < 0.001$ (two-way repeated measures (RM) ANOVA followed by Bonferroni's multiple comparisons test). For detailed statistical information, see Supplementary Tables 2–4.

compared with scrambled ODN (SCR-ODN). PSD-95 AS-ODN blocked memory reinstatement given 7 days after training [presentation of context (T1) followed, 2 days later, by a reminder shock (RS)], a protocol that significantly re-instates the apparently forgotten memory of SCR-ODN-injected rats (T2, Fig. 2c). The AS-ODN-injected rats learned the IA task when retrained (Tr) upon entering the shock compartment at T2, demonstrating that the AS-ODN had not functionally impaired the hippocampus (T3, Fig. 2c). Conversely, similar PSD-95 AS-ODN injections in rats trained at PN24 had no effect on memory retention (Fig. 2d), suggesting that at this age, de novo hippocampal PSD-95 is not required to form IA memory, hence indicating the existence of differential molecular regulations in infantile learning, when memory abilities are in an early phase of development.

Synapse maturation is marked by incorporation of GluN2A-containing NMDA receptors and an increase in the AMPA/NMDA current ratio. Training at PN17 increases the levels of GluN2A more abundantly than those of GluN2B causing a significant increase in the ratio GluN2A/GluN2B[22]. Here, we investigated whether AMPA receptor synaptic function change in response to IA learning at PN17 and/or PN24. To this end, we generated acute hippocampal slices from rats trained at PN17 or PN24, euthanized 24 h later, and compared them to slices obtained from naive rats at matched ages.

We performed whole-cell current clamp recordings from pyramidal neurons in the granule cell layer of area CA1 of the dHC (Fig. 3a). For each cell, we first recorded active and passive membrane properties (Fig. 3a). We observed no effect of training on firing rates (as a function of injected current), resting membrane potential or membrane resistance at PN17 or PN24. This suggests that intrinsic excitability of CA1 hippocampal pyramidal cells did not change in response to training.

Training at PN17, but not at PN24, caused a significant increase in the amplitude of evoked AMPA receptor excitatory postsynaptic potentials (EPSPs) and a significant decrease in EPSP decay times (Fig. 3b, c). These learning-induced changes resembled the normal maturation that occurred in naive animals from PN17 to PN24. Training at PN17 or PN24 did not affect paired-pulse ratios, a proxy for presynaptic release (Fig. 3d), although paired-pulse facilitation was stronger in the younger animals. Together, these data indicated that training at PN17 induces changes in postsynaptic AMPA receptor magnitude and kinetics that are typical of excitatory synapse maturation responses. These learning-induced changes in AMPA receptor responses required PSD-95. In fact, bilateral dHC injections of PSD-95 AS-ODN, compared with SCR-ODN, blocked the increase in the amplitude of evoked AMPA receptor EPSPs (Fig. 3f) and the decrease in EPSP decay times (Fig. 3g) without affecting the intrinsic excitability of CA1 hippocampal pyramidal cells (Fig. 3e) or the paired-pulse ratios (Fig. 3h).

Collectively, these data led us to conclude that, upon learning at PN17, the infantile hippocampus undergoes a long-lasting neuronal activation that follows a distinct kinetic relative to that observed in PN24 and adult hippocampus. This activation is accompanied by BDNF- and PSD-95-dependent excitatory synapse formation and/or maturation, which is necessary for long-term storage of infantile memory.

**Memory abilities mature in response to infantile learning**. Given that training at PN17 results in the hippocampal increase in excitatory synapse formation/maturation proteins, we hypothesized that a second learning event given 24 h or 48 h after the first one (i.e., at a time when most of the training-induced hippocampal molecular changes are significantly elevated) might capitalize on the induced biological maturation of the hippocampal memory system, hence promoting the ability to express long-lasting memories. As long-term memory expression is typical of the mature hippocampal memory systems, we refer to this ability as functional competence.

We trained rats in IA at PN17, and then administered a second IA training trial 24 h or 48 h later (Fig. 4a). To account for developmental age, we also tested the effects of a single training given at PN19 (Fig. 4b). As expected, the first training given at either PN17 or at PN19 resulted in rapid forgetting; when the rats were tested a day later they had latencies that were not above the acquisition level (Fig. 4a, b). However, the rats that received a second training trial expressed a robust and long-lasting IA latency (Fig. 4a, b). Naive and shock-only rats, used as controls, did not exhibit any significant latency above the acquisition level, excluding the contribution of nonspecific responses (Fig. 4a). We concluded that a second learning presented 24 h or 48 h after the first one, during a temporal window of molecular maturation evoked by a first learning experience, results in the acquisition of functional competence.

To refine the temporal window required for the functional maturation, and to control for the effects of receiving two training trials, as well as for developmental age, we tested memory retention after two training trials given 6 h apart at PN17 (2 × PN17) or at PN18 (2 × PN18). In contrast to the PN17 + PN18 group, which expressed long-term IA memory, both 2 × PN17 and 2 × PN18 had no significant memory 1 day later as well as at PN24 when they were re-tested (Fig. 4c). In agreement with the slow and persistent molecular maturation observed after learning at PN17, these data indicate that >6 h from the first learning are needed to allow for sufficient biological maturation that can be captured by the second learning to promote maturation of functional competence.

Moreover, to assess the persistence of the memory induced by two training trials, one at PN17 and the second at PN18, we tested the memory 29 days later (at PN47). As shown in Fig. 4d, memory expression was significant at this remote time point, although it was substantially decreased compared with memory tested at 1 or 6 days after training (Fig. 4a). Thus, repeated spaced learning in infancy leads to memory competence, and this mature memory, like that formed in adulthood, is long lasting.

To prove whether the changes in excitatory synapse proteins elicited by the first learning are necessary for the second learning event to capture the maturation and promote functional competence, we blocked the induction of PSD-95 following the first learning at PN17 and tested whether memory retention after the second learning trial at PN18 remained immature, that is, rapidly decaying. As shown in Fig. 4e, in comparison with SCR-ODN, bilateral hippocampal injections of PSD-95 AS-ODN after training at PN17 prevented the ability of the second training given at PN18 to produce long-term memory, tested 1 day (T1) and 6 days (T2) later. The AS-ODN-injected rats learned the IA task when retrained (Tr) upon entering the shock compartment at T2, demonstrating that the AS had not functionally impaired the hippocampus (T3, Fig. 4e).

These data showed that increase in the excitatory synapse marker PSD-95 induced by learning at PN17 is required for a second episodic learning to mature the functional competence of the long-term memory system. The PSD-95-dependent changes require >6 h in order to allow for sufficient biological maturation captured by the second learning event.

**Learning in infancy shapes memory abilities**. The hippocampal memory system processes aversive and nonaversive types of contextual, spatial, and spatiotemporal information[39]. Here, we

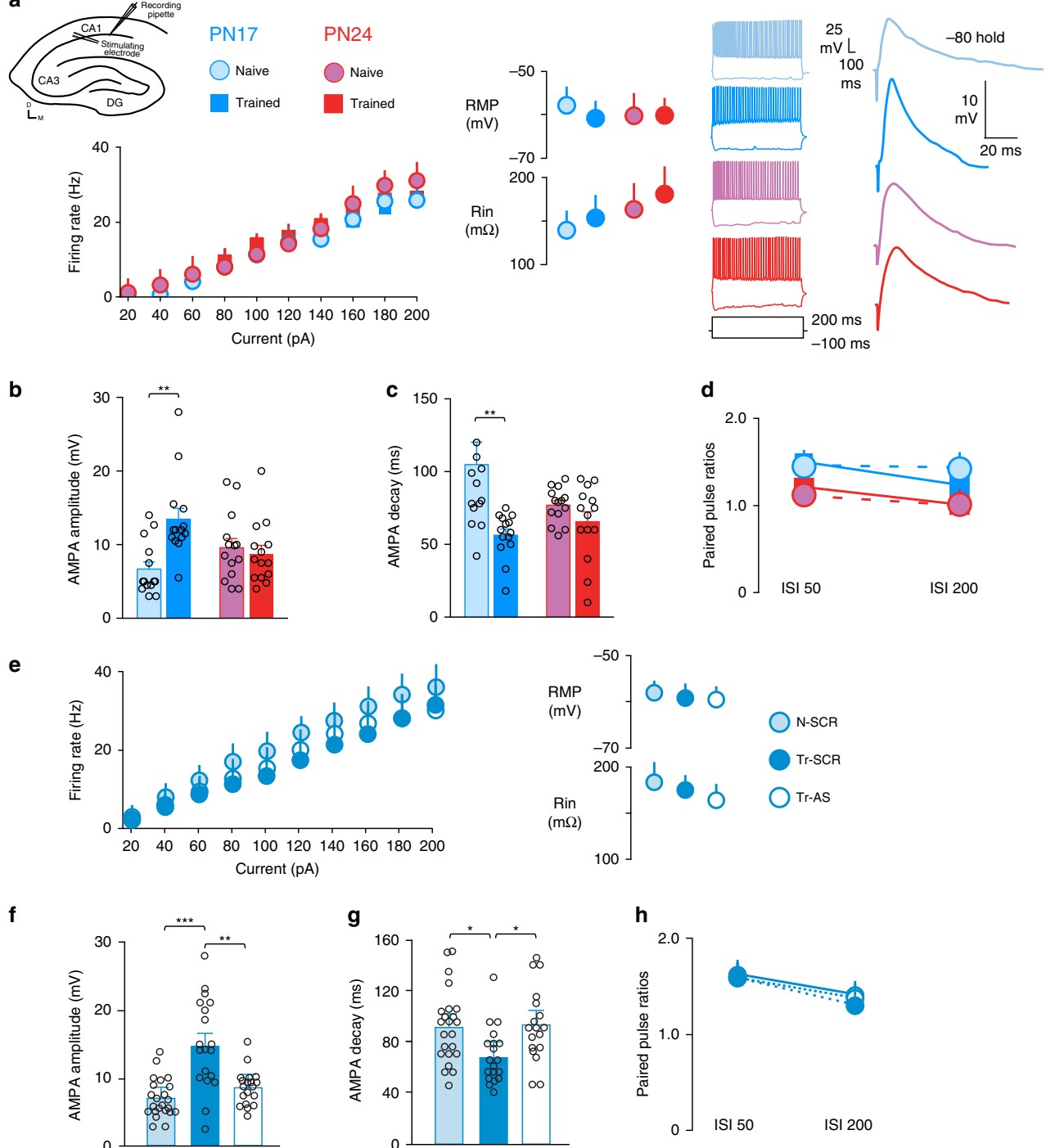

**Fig. 3 Learning-induced changes in AMPA receptor responses at PN17 require the expression of PSD-95. a** Diagram shows where whole-cell recordings were carried out in hippocampal slice preparations, and where a biphasic stimulator was placed to evoke AMPAergic responses. From left to right: intrinsic excitability data showing the effect of training at PN17 or PN24 on firing rate, resting membrane potential and membrane input resistance, representative traces of hippocampal pyramidal neurons following positive and negative current injection into the cell, and representative traces from the same cells showing evoked AMPA receptor potentials. **b, c** AMPA receptor potential amplitude (**b**) and decay time (**c**) after training at PN17 and PN24. **d** Paired-pulse ratios after training at PN17 and PN24. Sample size for **a–d**, each group, n = 14 cells obtained from three male rats per age. **P < 0.01 (two-way ANOVA followed by Bonferroni's multiple comparisons test). **e** Recordings were obtained from naive (N) and trained (Tr) rats, which received two hippocampal injections of scrambled (SCR) or PSD-95-antisense ODN (AS), the first immediately after training at PN17 and the second 6 h later. Rats were euthanized 24 h after training (i.e., PN18). These recordings were conducted in slices immediately in front of and behind the slice containing the cannula implant site. From left to right: Intrinsic excitability data showing the effect of training at PN17 and PSD-95 AS-ODN injections on firing rate, resting membrane potential, and membrane input resistance. **f, g** The effect of PSD-95 AS-ODN or SCR-ODN injections following training at PN17 on AMPA receptor potential amplitude (**f**) and decay time (**g**). **h** The effect of PSD-95 AS-ODN or SCR-ODN injections following training at PN17 on paired-pulse ratios. Sample size for **e–h**, N-SCR, n = 23 cells; Tr-SCR, n = 19 cells; Tr-AS, n = 18 cells; obtained from three male rats per group. *P < 0.05, **P < 0.01, ***P < 0.01 (one-way ANOVA followed by Tukey's multiple comparisons test). For detailed statistical information, see Supplementary Tables 5–6.

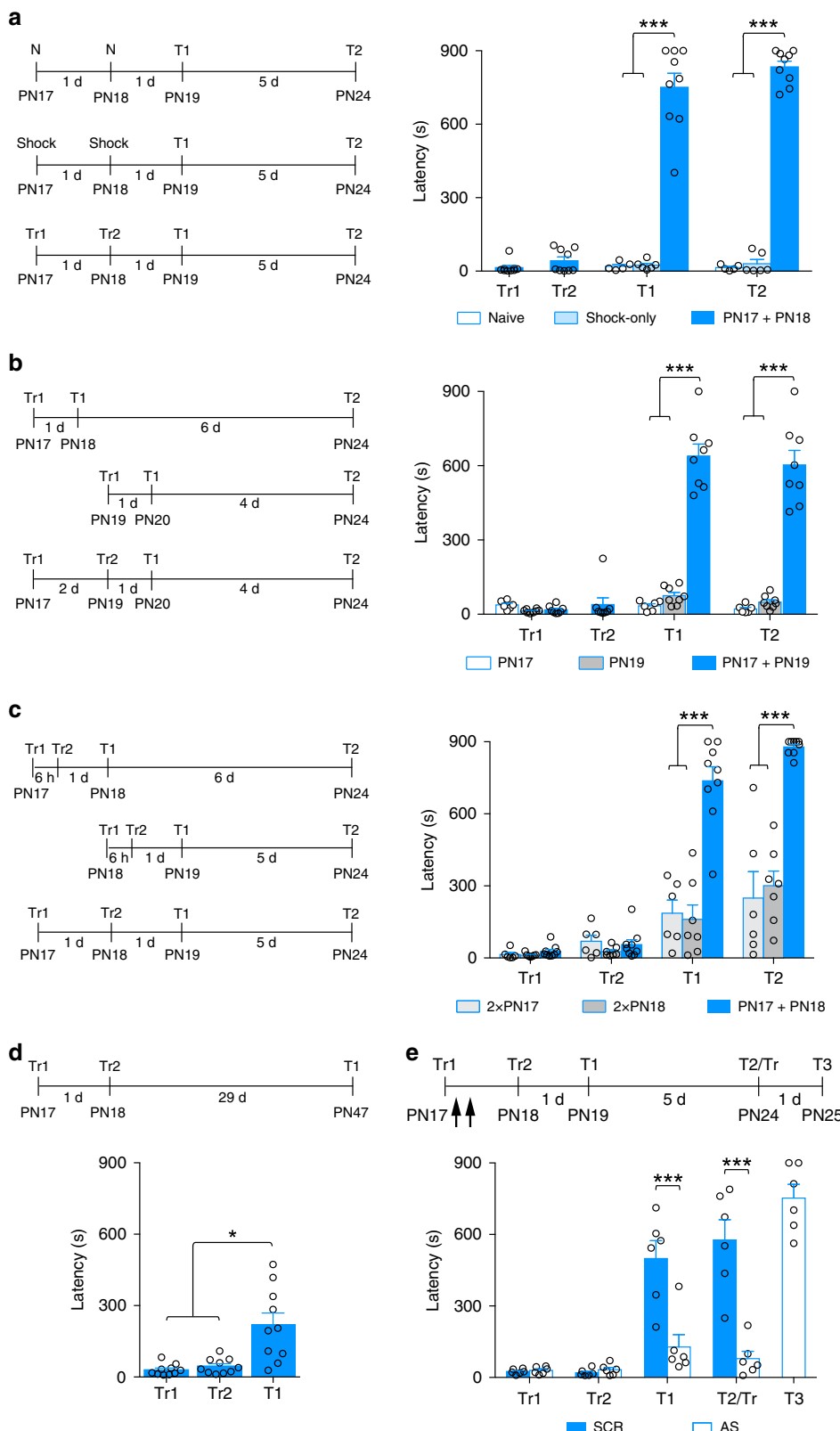

asked if the functional maturation of the hippocampal memory system is experience-selective. In other words, we tested whether a given type of hippocampal learning (e.g., IA) matures functional competence only for that type of experience or whether it develops the functional competence of the hippocampal memory system as a whole, hence of other types of hippocampus-dependent learning. Toward this end, we assessed whether contextual aversive learning (i.e., IA) matures only IA memories (and whether this maturation is context-specific) or instead also matures spatial discrimination memory competence (nOL). In addition, we asked the same question using first the nOL and then the IA paradigm.

**Fig. 4 Two learning events spaced by 24 h or 48 h result in long-term memory expression. a** Rats were either left in their home cage (naive, N; $n = 5$), exposed to a shock-only protocol ($n = 6$) or received two training trials, one at PN17 (Tr1) and the second at PN18 (Tr2) (PN17 + PN18; $n = 9$). Rats were tested 1 day after the second training (T1) and again 5 days later (T2) or at matched time points for the other 2 groups. **b** Rats received either one training trial (Tr1) at PN17 ($n = 6$) or at PN19 ($n = 8$), or two training trials, one at PN17 (Tr1) and the second at PN19 (Tr2) (PN17 + PN19; $n = 8$). Rats were tested 1 day after the last training (T1) and again at PN24 (T2). **c** Rats received two training trials, one at PN17 and the second 6 h later (2xPN17; $n = 6$), or one at PN18 and the second 6 h later (2xPN18; $n = 7$), or one training at PN17 and the second at PN18 (PN17 + PN18; $n = 9$). The animals were tested 1 day after the last training (T1) and again at PN24 (T2). **d** Rats ($n = 10$) were trained at PN17 (Tr1), and again at PN18 (Tr2), and then tested 29 days later (T1). **e** Rats received two training trials, one at PN17 (Tr1) and the second at PN18 (Tr2). Rats were injected twice (black arrows) in the dHC with either SCR ($n = 6$) or AS ($n = 6$) against PSD-95 immediately after training at PN17 and 6 h later. Rats were tested at PN19 (T1), and at PN24 (T2). At T2, upon entering the shock compartment, rats were trained again (Tr) and tested 1 day later (T3). Memory retention is expressed as mean latency ± s.e.m. (in seconds, s). *$P < 0.05$ (one-way RM ANOVA followed by Tukey's multiple comparisons test); ***$P < 0.001$ (two-way RM ANOVA followed by Bonferroni's multiple comparisons test). For detailed statistical information, see Supplementary Table 7.

We trained rats in IA at PN17 in context A, and again 24 h later, at PN18, either in the same context (context A) or—to assess whether there was a generalization and/or transfer effect to a different context—in a distinct context (context B). Memory was tested in both contexts 24 h later, then again 5 days later. Two training trials in context A resulted in long-term memory for context A but not context B (Fig. 5a), indicating that the long-lasting memory formed was context-specific and did not generalize to other contexts. Training in context A followed by training in context B produced long-lasting memory of both contexts (Fig. 5b), indicating that the maturation produced by an IA training in one context was captured by a similar type of learning in a different context. Hence, infantile IA learning promotes the competence for the IA memory domain.

These data indicated that the maturation of the abilities to express long-term memory, and hence functional competence, transfers to similar learning domains (i.e., IA experienced in two distinct contexts), although the stored memory remains specific for the experiences encountered and does not generalized to non-experienced representations.

To determine whether maturation by two spaced training trials also occurs in nonaversive types of hippocampal memory, we employed nOL in rats. Previously, we showed that infant rats trained at PN17 in nOL express memory retention 1 min after training, but not 2 h later, recapitulating the rapid forgetting of infantile memories in nonaversive tasks[40]. Here, we found that two nOL training trials spaced by 24 h—in the same context, with the same objects in the same positions—yielded significant memory 2 h later (Fig. 5c). Similar to the results obtained with IA, rats trained at PN17 and PN18 with two different object pairs in two different placements also acquired the ability to prolong memory expression of the moved object, indicating that maturation induced by one experience is transferred to another experience of the same domain (Fig. 5c).

However, when two distinct hippocampal learning domains, IA and nOL, were given 24 h apart in either order (i.e., IA followed by nOL (IA + nOL) or nOL followed by IA (nOL + IA), the maturation of functional competence was not observed. In fact, rats trained in IA + nOL did not exhibit any nOL memory 2 h later, suggesting that IA memory did not facilitate the behavioral maturation of nOL memory competence (Fig. 5d). These rats were able to reinstate IA memory after T1 + RS given 7 days later, indicating that the nOL training did not interfere with the original IA memory (Fig. 5d).

To increase the possibility of interaction, we performed the nOL + IA paradigm under conditions of increased maturation of nOL, i.e., in the presence of BDNF. We showed previously that BDNF injection in the hippocampus immediately after nOL promotes memory retention 2 h after training[40]. Rats trained in nOL at PN17 received a bilateral hippocampal injection immediately after training, and IA training a day later. No IA

memory retention above acquisition level was observed 1 or 7 days after IA training, suggesting that maturation of one memory competence is not transferred to another type of hippocampal memory competence. Nevertheless, the rats were able to express IA memory after the T2 + RS, indicating that a matured nOL memory did not interfere with IA acquisition or infantile memory formation (Fig. 5e).

We concluded that functional maturation of the hippocampal memory system is experience-specific, and that experience-dependent maturation transfers to similar learning domains, but not to distinct ones.

This conclusion implies that similar learning experiences, but not distinct ones, mature memory competence because they recruit overlapping hippocampal cell ensembles. If this was true, blocking the cellular activation induced by the first learning should prevent the acquisition of the functional competence seen after a second, similar, learning experience. To test this hypothesis, we employed c-Fos AS-ODN double injections (immediately post-training and 6 h later) after IA learning at PN17, which, as expected, blunted the induction of c-Fos following training (Fig. 6a). This knockdown prevented long-term memory expression following the second learning at PN18 (Fig. 6b), indicating that the c-Fos-mediated cell ensemble activation induced by the first learning is necessary for enabling functional competence. In other words, a second, similar learning event must capture an overlapping c-Fos cell ensemble activated by the first learning to promote functional competence. The c-Fos AS-ODN did not disrupt the ability of the rats to re-learned IA when retrained (Tr) upon entering the shock compartment at T2 (T3, Fig. 6b), indicating that c-Fos AS-ODN injections did not disrupt hippocampal functions.

**Artificial maturation of functional competence.** We next tested two questions: first, is artificial re-stimulation of the cell ensemble activated during learning sufficient to promote maturation of memory expression? Second, if so, is the artificially created maturation selective for the initial experience? To this end, we employed c-fos-htTA/tetO-hM3Dq double-transgenic mice[41], which express the hM3Dq protein under the regulation of the c-fos promoter in a doxycycline (Dox)-dependent manner. Clozapine-N-oxide (CNO) injection into these mice stimulates cells expressing hM3Dq induced by the c-fos promoter activated at training. c-fos-htTA and tetO-hM3Dq mice were used as controls. Consistent with previous studies[23,42], mice trained in contextual-fear conditioning (CFC) exhibited rapid forgetting: they had a robust memory of the context 1 day after training, but memory retention returned to baseline levels 7 days after training (T2; Fig. 7a). Similar to rat IA, CFC in PN17 mice induced a gradual, lasting upregulation of c-Fos in the dHC, which peaked 24 h after training and returned to baseline a day later (Fig. 7b).

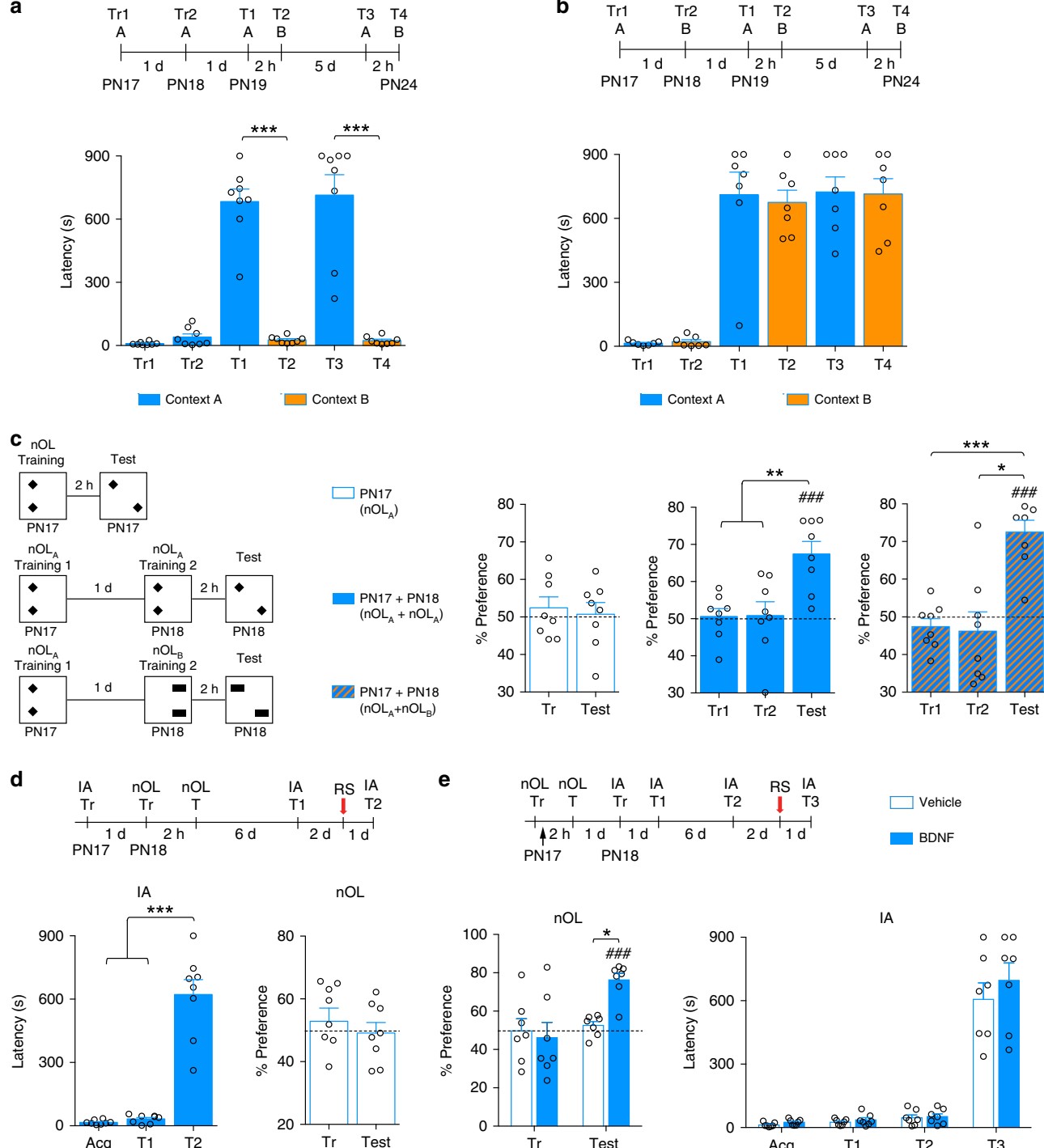

**Fig. 5 Maturation of memory competence is selective for the type of learning domain. a, b** Rats were trained (Tr1) in IA at PN17 in context A and retrained (Tr2) at PN18 in the same context (*n* = 8) (**a**) or in a different context, context B (*n* = 7) (**b**). Rats were tested 1 day later in context A (T1) and 2 h later in context B (T2). Five days later, they were again tested in context A (T3), and 2 h later in context B (T4). **c** Rats were trained in nOL following three schedules (*n* = 8 rats per schedule): (i) one training trial (Tr) at PN17; (ii) two training trials (Tr1 and Tr2) in the same context with the same object locations spaced by 1 day (nOL$_A$ + nOL$_A$), or (iii) two training trials spaced by 1 day, but with the second trial presenting a different pair of objects in a different location (nOL$_A$ + nOL$_B$). All rats were tested 2 h after the last training (Test). **d** Rats (*n* = 8) were trained in IA at PN17, and then trained in nOL at PN18. Rats were tested in nOL 2 h after nOL training (nOL test, T), and in IA 7 days after IA training (IA T1) then 1 day (IA T2) following the reminder shock (RS). **e** Rats (*n* = 7 per group) were trained in nOL at PN17, bilaterally injected in the dHC with BDNF immediately after training (black arrow), and tested in nOL 2 h afterward (nOL test, T). Then, the rats were trained in IA at PN18, and tested 1 day (IA T1) and 7 days later (IA T2), then again 1 day (IA T3) following the RS. *$P$ < 0.05, **$P$ < 0.01, ***$P$ < 0.001 (one-way RM ANOVA followed by Tukey's multiple comparisons test). ###$P$ < 0.001 significance for one-sample *t* tests comparing each group to chance performance (50%). IA memory retention is expressed as mean latency ± s.e.m. (in seconds, s). nOL memory retention is expressed as % time spent exploring the moved object ± s.e.m. For detailed statistical information, see Supplementary Tables 8–11.

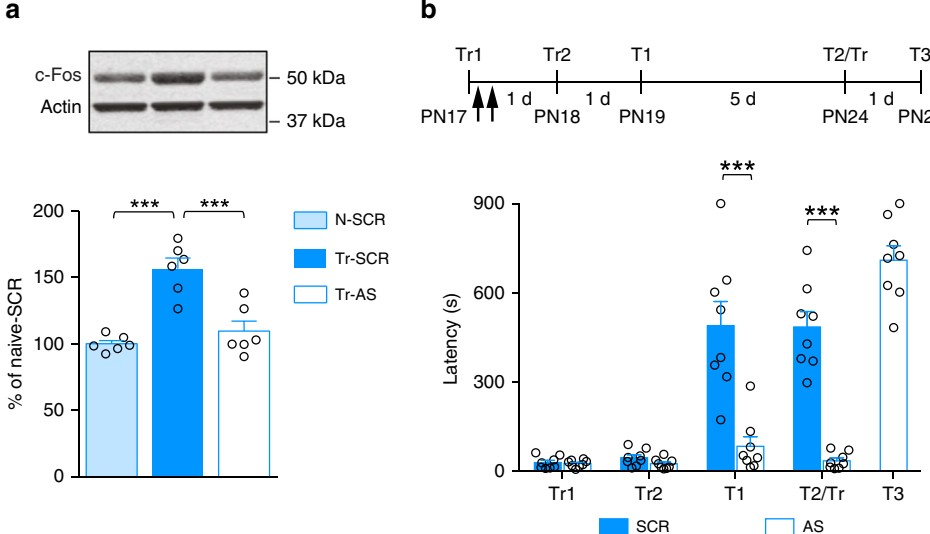

**Fig. 6 Blocking c-Fos induction after the first learning experience precludes the second learning from promoting functional competence.**
**a** Representative examples and bar graphs showing c-Fos densitometric western blot analyses of dHC extracts obtained from naive (N) and trained (Tr) rats, which received two hippocampal injections of scrambled (SCR) ($n = 6$ rats per group) or c-Fos-antisense ODN (AS) ($n = 6$ rats), the first immediately after training at PN17 and the second 6 h later. Protein extracts were obtained 24 h after training. Data are expressed as mean percentage ± s.e.m. of the naive group injected with SCR-ODN. ***$P < 0.001$ (one-way ANOVA followed by Tukey's multiple comparisons test). **b** Rats received two training trials, one at PN17 (Tr1) and the second at PN18 (Tr2). Rats were injected twice (black arrows) in the dHC with either SCR-ODN ($n = 8$ rats) or AS-ODN ($n = 8$ rats) against c-Fos immediately after training at PN17 and 6 h later. Rats were tested at PN19 (T1), and at PN24 (T2). At T2, upon entering the shock compartment, rats were trained again (Tr) and tested 1 day later (T3). Memory retention is expressed as mean latency ± s.e.m. (in seconds, s). ***$P < 0.001$ (two-way RM ANOVA followed by Bonferroni's multiple comparisons test). For detailed statistical information, see Supplementary Tables 12–13.

When c-fos-htTA/tetO-hM3Dq mice trained at PN17 received a systemic injection of CNO 30 min before a first test given at 7 days after training, they exhibit significant CFC memory retention 7 days after training (T1) relative to control littermates carrying either only c-fos-htTA or only tetO-hM3Dq, and were injected with CNO at the same time (Fig. 7c). Memory retention remained elevated 7 days later (T2). These data indicate that stimulating the cellular network marked by activation upon learning is sufficient to promote the maturation of functional competence, underscoring the importance of a specific cellular representation in the hippocampus that matures in response to learning.

We then investigated whether a behavioral reminder of the original experience, instead of artificial stimulation of the cellular network, would reinstate CFC memory in mice. For this purpose, we employed a saving protocol, in which mice were trained at PN17 and then exposed to a single low-intensity footshock in the training context 7 days after training. These mice reinstated significant memory retention 1 day after the saving protocol (T3) relative to mice that did not receive training at PN17 and were exposed only to the saving protocol (Fig. 7d). These data confirm that infantile memory, apparently forgotten, is not lost and can re-emerge at later ages (Fig. 7c, d).

To confirm that the maturation of the memory expression is also selective for the learning encountered in mice, we investigated the influence of CFC and nOL learning given 1 day apart, similar to what was done in rats with IA and nOL. We found that mice trained with CFC given once at PN17 and then again at PN18 (PN17 + PN18) had significant memory retention both 1 and 7 days after training (Fig. 8a), whereas a single training trial at PN17 was not sufficient to elicit memory retention 7 days after training (Fig. 8a). PN17 mice trained in nOL exhibited significant memory retention 1 min after training, but not 2 h later (Fig. 8b). Similarly to rats, if these mice were exposed to CFC a day later, they failed to exhibit any memory

retention of CFC 6 days later, leading us to conclude that nOL learning did not affect the ability to mature CFC memory (Fig. 8c).

Finally, we tested whether the artificial stimulation of the cell ensembles activated upon learning, like repetition of the same behavioral experience, produces an experience-specific maturation of functional competence. Hence, we tested whether artificial reactivation of a CFC learning affects the long-term memory retention of nOL. Mice trained in CFC at PN17 were then injected with CNO 30 min before a nOL trial given at PN18 resulted in a significant CFC memory retention at both 2 and 7 days after the CFC training (Fig. 8d). However, these mice failed to exhibit any nOL memory retention at 2 h after nOL training (Fig. 8d), demonstrating that artificial reactivation of the cell ensembles activated upon CFC learning promotes long-term memory expression, hence functional competence, for CFC, but this functional maturation does not transfer to nOL memory.

We concluded that a learning experience in infancy activates a specific cellular ensemble representation that supports selective functional maturation of that learning domain, but not of distinct types of hippocampal learning domains. Hence, the maturation of abilities underlying hippocampal memory competence is selectively shaped according to the experiences learned.

## Discussion
Our results showed that infantile learning, during the period of rapid "forgetting", produces unique molecular, cellular, and synaptic maturation changes. These include slow and persistent inductions of the IEGs Arc, c-Fos, and Zif268, of structural excitatory synaptic proteins PSD-95 and synaptophysin, and of phosphorylation of AMPA receptors at Ser-831 and Ser-845. The temporal kinetics of these molecular changes appeared to be strikingly distinct from those of both PN24 and adult rats, which showed IEG rapid induction and return to baseline levels within

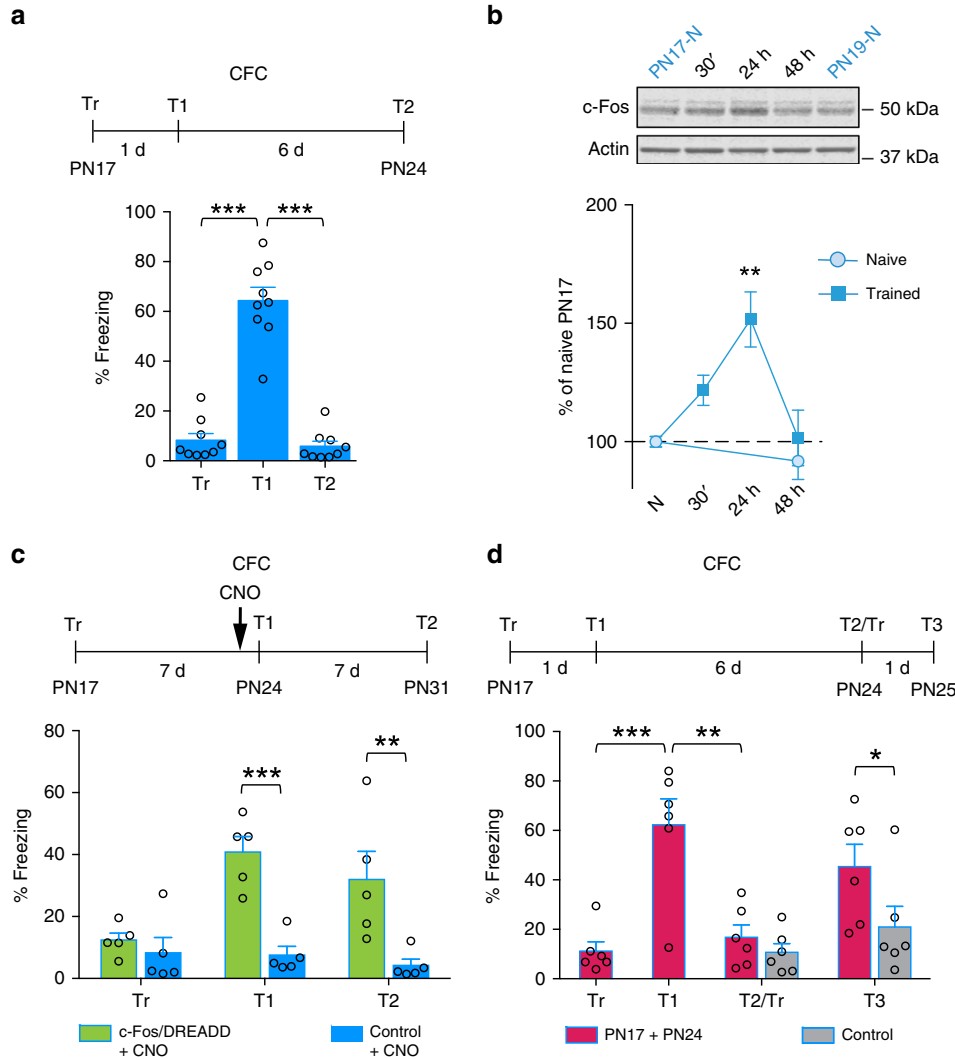

**Fig. 7 Chemogenetic reactivation of the cellular networks activated at learning, as well as relearning, matures functional competence. a** Mice ($n = 9$) were trained in CFC at PN17 and tested 1 day (T1) and 7 days later (T2). ***$P < 0.001$ (one-way RM ANOVA followed by Tukey's multiple comparisons test). **b** Densitometry of c-Fos western blot analysis obtained from dHC whole-protein extracts from mice trained in CFC at PN17 and euthanized 30 min, 24 h, or 48 h after training (naive PN17, $n = 5$; 30 min, $n = 5$; 24 h, $n = 6$; 48 h, $n = 5$; naive PN19, $n = 3$). To account for developmental differences, two groups of naive (N) rats were used (PN17 and PN19). **$P < 0.01$ versus PN17-N (one-way ANOVA followed by Dunnett's multiple comparisons test). Actin was used as a loading control. Data are expressed as mean percentage ± s.e.m. of the PN17 naive group. **c** Mice ($n = 5$ per group) were trained in CFC at PN17 and injected with CNO 30 min before being tested 7 days after training (T1). Mice were tested again 7 days later (T2). **$P < 0.01$; ***$P < 0.001$ (two-way RM ANOVA followed by Bonferroni's multiple comparisons test). **d** Mice ($n = 6$ per group) were trained at PN17 or remained in the home cage (control), and were tested 1 day (T1) and 7 days later (T2/Tr). At T2, mice received a single low-intensity shock (0.3 mA) in the training context (saving), and were tested 1 day later (T3). *$P < 0.05$; **$P < 0.01$; ***$P < 0.001$ (two-way RM ANOVA followed by Bonferroni's multiple comparisons test). CFC memory retention is expressed as mean % freezing ± s.e.m. For detailed statistical information, see Supplementary Tables 14–17.

2–3 h after training, typical of functionally mature adult systems[25–27], as well as no significant changes in levels of synaptic proteins and AMPA receptor phosphorylation at Ser-831 and 845. Moreover, the BDNF-dependent PSD-95 upregulation following infantile training, which was necessary for both AMPA receptor response maturation and long-term memory storage, led us to conclude that infantile learning instructs formation and/or maturation of excitatory synapses, which are critical for long-term memory storage.

These distinctive molecular regulations of the infant hippocampus following learning suggest that the hippocampus-dependent memory system at this age, compared with later developmental stages, is biologically different. The bases for these unique biological regulations remain to be understood, and, at present, we can only offer some speculations. For example, the gradual and sustained

upregulation of IEGs following training at PN17 may reflect a higher level of hippocampal synaptic plasticity and neuronal excitability at this age compared with later ages[43–46]. This hypothesis is in line with the observation that basal expression of IEGs and of synaptic plasticity and neuronal excitability markers, such as cAMP response element-binding protein (CREB)[47] and NMDA receptor GluN2B subunit[48–50], is higher in the dHC of infant rats compared with adults[24]. Moreover, the prolonged IEG induction in the infant brain could result from a lower GABAergic inhibition[24,51], as suggested by greater IEG expression in the lateral amygdala following chemogenetic inhibition of parvalbumin-expressing interneurons during memory formation in adult mice[52].

The mechanisms by which learning at PN17, but not at PN24, lead to PSD-95-dependent changes in AMPA receptor responses in the hippocampus are also unknown. The two ages may have

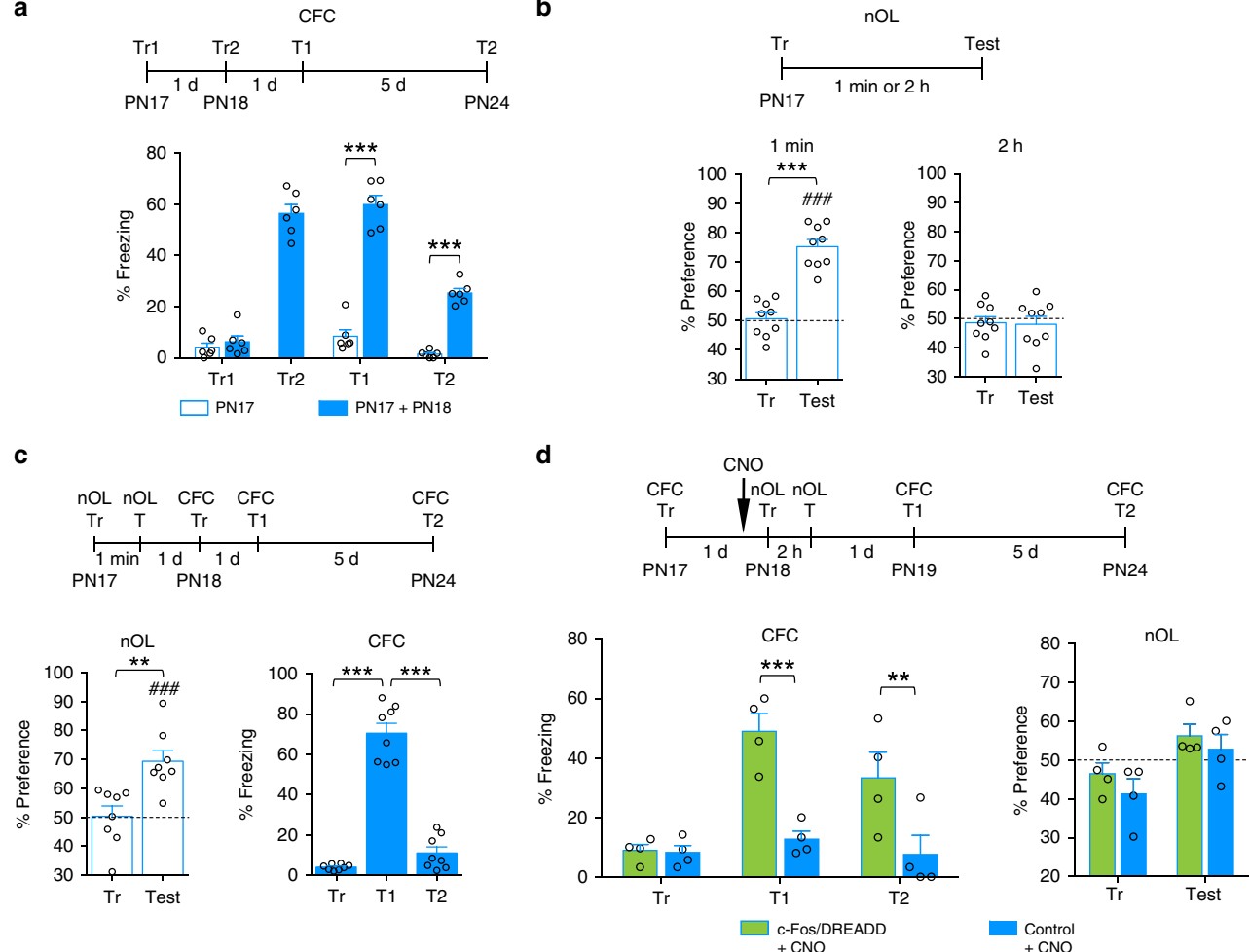

**Fig. 8 Acquisition of functional competence in mice is learning-selective. a** Mice ($n = 6$ per group) received two CFC trainings, one at PN17 and the second at PN18. They were tested 1 day after the last training (T1) and again 5 days later (T2). ***$P < 0.001$ (two-way RM ANOVA followed by Bonferroni's multiple comparisons test). **b** Mice ($n = 9$ per retention time) were trained in nOL at PN17 and tested 1 min or 2 h after training. ***$P < 0.001$ (two-tailed paired Student's $t$ test). ###$P < 0.001$ significance for one-sample $t$ tests comparing each group to chance performance (50%). **c** Mice ($n = 8$) were trained in nOL at PN17, and tested for nOL 1 min later (nOL test, T). One day later, all mice were trained in CFC (PN18) and tested 1 day (T1) and 5 days (T2) later. **$P < 0.01$ (two-tailed paired Student's $t$ test); ###$P < 0.001$ significance for one-sample $t$ tests comparing each group to chance performance (50%). ***$P < 0.001$ (one-way RM ANOVA followed by Tukey's multiple comparisons test). **d** Mice ($n = 4$ per group) were trained in CFC at PN17 and injected with CNO 30 min before training at PN18 in nOL. Mice were tested for nOL memory 2 h later, then tested in CFC 1 day later (T1), and again 5 days later (T2). **$P < 0.01$; ***$P < 0.001$ (two-way RM ANOVA followed by Bonferroni's multiple comparisons test). CFC memory retention is expressed as mean % freezing ± s.e.m. nOL memory retention is expressed as % time spent exploring the moved object ± s.e.m. For detailed statistical information, see Supplementary Tables 18–21.

different rates or regulations of receptor trafficking, of synaptic levels of AMPA receptor subunits (GluA1-4), and/or different learning-induced post-translational modifications. Based on our data, we speculate that infantile learning, via increased phosphorylation of GluA1 subunits at Ser-831 and Ser-845, may lead to changes in the AMPA receptor biophysical properties and synaptic expression[28,30], which could result in increase in AMPA receptor EPSP amplitude and decrease in decay times. Finally, learning at PN17 may involve other types of changes to AMPA receptors, which still remain to be identified.

The differential biological changes in markers of synapse formation/maturation found in the hippocampus at PN17 compared with PN24 are in agreement with and expands on our previous findings[22,40] showing that learning in infancy engages mechanisms typically recruited during critical periods[32]. Collectively, these data further support our hypothesis of the existence of a developmental temporal window during which

maturation of learning and memory abilities and its underlying biology occur[10].

The learning-induced biological maturation suggests that building and stabilizing mature cellular and synaptic networks, and perhaps also the development and growth of brain structures, occur selectively according to the experience encountered. Furthermore, the observed slow kinetics of molecular and cellular changes evoked by infantile learning provides a possible mechanistic link for the protracted development of the hippocampus-dependent learning and memory system[53]. Our data suggest that this system builds slowly over experiences during the infantile period, a conclusion in agreement with structural and functional changes found in humans[54], and the important role of experience deprivation as well as the type of experience encountered by children in early life[55,56].

Early observations in 1960s pioneered by Rosenzweig, Bennett, and Diamond et al.[57] led to the groundbreaking demonstration

that training or differential experience changes the neurochemistry and neuroanatomy of the rat cerebral cortex leading to modifications in cortical thickness, size of synaptic contacts, number of dendritic spines, and dendritic branching. Although some behavioral selective correlations were documented[58], most of those studies assessed the effect of environmental enriched experience, hence of a variety of stimuli, and reported that enrichment can improve performance on several behaviors accompanied by a variety of biological changes[57,59,60]. Other early studies reported an increase in the weight of the hippocampus in birds storing and retrieving food, suggesting that anatomical changes may occur selectively in brain regions implicated in the type of experience given[61]. Despite these pioneering findings, until now, because of technology limitations, it remained unclear whether the modifications produced by developmental experience structurally shape the brain in a selective manner. Our results proving a critical impact of the specificity of experiences during early development in maturing molecular, circuital, and behavioral functions of the hippocampus-dependent memory system, provide evidence that the stimuli that the brain receives in early life shape the developmental biology and abilities long term. They also offer a potential explanation for why early-life experiences have repercussions throughout life.

Because the hippocampus is part of an integrated memory system, we speculate that the observed cellular maturation is not limited to the hippocampus but extend to the whole hippocampus-dependent memory system.

In addition, our findings showing that memory competence acquired by repeated trials is long lasting and detectable at remote time points, suggest that infantile memories undergo a systems-level reorganization. However, given the immaturity of the infant hippocampal memory system, we cannot exclude that system consolidation in infants is different than in adults.

Our chemogenetic stimulation experiments employed intra-peritoneal CNO administration because cannulae implantation in infant mice is technically challenging due to the small size of the mouse brain at this age. Because CNO systemic administration to the c-fos-htTA/tetO-hM3Dq mice triggers a forebrain-wide activation of the hM3Dq receptors[41], our data cannot exclude the possibility that, in addition to the hippocampus, other forebrain regions were activated and contributed to the acquisition of functional competence. Furthermore, the specificity of systemically administered CNO and/or its metabolites have been debated[62]. Therefore, to limit the potential side effects of systemic CNO injection, we employed a low CNO dose (i.e., 0.5 mg/kg), which had been previously found effective in activating hM3Dq receptors in the hippocampus[41,63]. Finally, to control for CNO-induced non-memory-related effects, we injected the same dose of CNO to control mice that did not express the hM3Dq receptors.

Our results also imply that specific experiences during the infantile developmental period make a major contribution to individual differences in learning and memory abilities. Although all individuals are exposed to general learning of facts, people, things, time, and spaces, and therefore must develop a wide range of abilities and competences processed by the hippocampal memory system, our results suggest that the individual history shapes the maturation of selective abilities. These conclusions may explain why early-life experiences influence the development of personality traits[64] and are in agreement with the idea of enduring individual effects of experiences consolidated during early childhood[65]. Therefore, we speculate that limited and/or selected experiences will build selected functional competences, whereas enriched, emotionally balanced, and diversified experiences will provide the greatest capacity for adaptive functional capabilities throughout life.

Memory development is important for thinking, future learning, planning, decision-making, problem solving, reflecting, imagining, and the overall capacity to form a sense of self. We suggest that regulation of infantile learning, especially during learning and memory critical periods, represents an extremely effective tool for preventing numerous psychopathologies.

## Methods

**Rats**. Seventeen- and 24-day-old male and female rats were obtained from E10/11 pregnant Long Evans female rats (Charles River Laboratories). Pre-weaning rats were housed with their littermates and mother in 30.80 cm × 40.60 cm × 22.23 -cm plastic cage, containing ALPHA-dri® bedding, under a 12 h/12 h light/dark cycle (light on at 7.00 a.m.) with food and water ad libitum. All experiments were carried out during the light cycle. Birth date was considered PN0, and the litters were culled to 10–12 pups (six males and six females, if possible) on PN1. Only one male and one female per litter were used in any experimental condition. After weaning (PN21), rats were group-housed (two to five per cage).

**Mice**. Seventeen-day-old male and female c-fos-htTA/tetO-hM3Dq mice were obtained by crossing c-fos-htTA mice (also referred to as B6.Cg-Tg(Fos-tTA, Fos-EGFP*)1Mmay/J mice; stock number 018306) and tetO-hM3Dq mice (also referred to as Tg(tetO-CHRM3*)1Blr/J; stock number 014093). To restrict activity-dependent labeling to training episodes, mice were raised on food containing 40 mg/kg doxycycline (DOX), and were taken off DOX 2 days before contextual-fear-conditioning (CFC) training. Mice were placed back on DOX food immediately after the CFC training, and remained on DOX for the remainder of the experiment. Mice were bred in the animal facilities at New York University under a 12 h/12 h light/dark cycle (light on at 07.00 a.m.) with food and water ad libitum. After weaning (PN21), mice were group-housed (two to four per cage) in transparent plastic cages (31 × 17 × 14 cm) with free access to food and water.

**Inhibitory avoidance (IA)**. Inhibitory avoidance (IA) was carried out in a chamber (Med Associates Inc., St. Albans, VT) consisted of a rectangular Perspex box divided into a safe compartment and a shock compartment (each 20.3 cm × 15.9 cm × 21.3 cm). The safe compartment was white and illuminated, and the shock compartment was black and dark. The apparatus was located in a sound-attenuated, non-illuminated room. During training sessions, each rat was placed in the safe compartment with its head facing away from the door. After 10 s (s), the door separating the compartments was automatically opened, allowing the rat access to the shock compartment. The door closed automatically when the rat entered the shock compartment with all four limbs, and a footshock (2 s, 1 mA) was administered. Footshocks were delivered to the grid floor of the shock chamber via a constant current scrambler circuit[22]. Animal remained in the dark compartment for an additional 10 s, and then returned to its home cage prior to testing for memory retention at designated time points. As controls, we used untrained animals (handled and then returned to their home cage, termed naive) and rats exposed to a footshock of the same intensity as that used in IA, but delivered immediately after they were placed on the grid, without presenting them with the IA context (shock only). This protocol does not induce any association between the context and the footshock[22]. Retention tests were performed by placing the rat back in the safe compartment, and measuring the latency to enter into the dark compartment. Footshocks were not administered during the retention tests (unless otherwise specified), and testing was terminated at 900 s. During retraining sessions, rats tested for memory retention received a footshock upon entering the dark compartment. Locomotor activity was measured in the IA chamber by automatically counting the number of times each rat crossed the invisible infrared light photosensor during training and testing. All behavioral tests were carried out blind to training and/or treatment conditions. For biochemical studies, rats were not tested for memory retention. Reminder footshocks (RS), with identical duration and intensity to those used in training, were administered in a novel, neutral chamber with transparent walls. Context generalization was tested in a new, modified IA box with a smooth plastic floor and decorated colorful walls, located in a different experimental room.

**Novel object location**. Animals (rats and mice) were habituated, trained, and tested in a square, open field (29 × 29 × 18 cm) with clear Plexiglas walls and floor located in a dim room[40]. Visual cues were provided within the box and on the walls of the room. The walls of the box were covered with white and black paper. One black and one white wall also contained symbols (circle and lines) to create four unique walls. Behavior was recorded with a video camera positioned ~1.5 m above the arena. Animals (rats or mice) were first habituated to the arena for 5 min for 2 consecutive days prior to training, approximately the same time each day in the mid-afternoon. Twenty-four hours after the last habituation session, each animal was returned to the arena for its training session. Training consisted of exposing the rats to two identical objects constructed from Mega Bloks® secured to the floor of the arena. Object sizes were age-appropriate, i.e., no taller than twice the size of the rat. Rats were initially placed facing a wall, away from the objects, and were allowed to explore the arena and objects for 5 min. At either 1 min or 2 h after

training, each animal was tested in the arena. During testing, one object remained in the same location as during training, whereas the second object had been moved to a novel location. Animals were placed in the arena facing the same direction as during training and were allowed to explore for 5 min. In all experiments, the placement of the object in the novel location was counterbalanced within each age group and time delay. The arena and objects were cleaned with 70% ethanol between sessions. Video files were coded and scrambled. An experimenter blind to treatment scored the total time that the rats spent actively exploring each object on each session. For both training and testing phases, exploration was defined as an active event in which the rat was pawing at, sniffing, or whisking with its snout directed at the object from a distance of less than ~1 cm. Sitting on or next to an object was not counted as active exploration. Memory was measured as the percentage of time spent exploring the object in the novel location compared with the stationary object.

**Contextual-fear conditioning.** Mice were conditioned in a fear-conditioning chamber, which consisted of a rectangular plexiglass box ($30.5 \times 24.1 \times 21.0$ cm) with a metal grid floor (Model ENV-008, Med Associates). During training, mice were placed in the CFC chamber for 120 s and then presented with three unsignaled footshocks (0.6 mA, 2 s duration, 1 min apart). Mice were removed from the chambers 1 min after the last shock. During the test, mice were placed in the chambers for 5 min. Testing occurred at the designated time points by placing mice in the chamber for 5 min[66]. No footshock was delivered during testing. All experiments were video recorded, and freezing (defined as lack of movement except for breathing) was scored by an experimenter blind to the treatment conditions. During the saving protocol, mice were placed in the CFC chamber for 120 s, and then presented with a single, mild footshock (0.3 mA, 2 s duration). Mice were removed from the chambers 1 min later. For chemogenetic memory reactivation experiments, mice were injected intraperitoneally with 0.5 mg/kg of clozapine-N-oxide (CNO) (Enzo Life Sciences, Farmingdale, NY, USA) or vehicle (10% DMSO) at the indicated times.

**Hippocampal cannula implants, injections, and ODN sequences.** On PN15 or PN22, pups were anesthetized with isoflurane mixed with oxygen. Stainless steel cannulas (26-gauge) were implanted bilaterally in the dHC (for PN15: −3.0 mm anterior, 2.2 mm lateral, and −2.3 mm ventral from bregma; for PN22: −3.4 mm anterior, 2.2 mm lateral, and −2.5 mm ventral from bregma) through holes drilled in the overlying skull. The cannulas were fixed to the skull with dental cement. After recovery from the surgery, pups were returned to the dam and littermates (for PN15) or their home cage (for PN22) for a 2-day recovery period prior to experimental manipulations. Hippocampal injections used a 33-gauge needle that extended 1 mm beyond the tip of the guide cannula, and was connected via polyethylene tubing to a Hamilton syringe. Injections were delivered using an infusion pump at a rate of 0.1 μl/min to deliver a total volume of 0.3 μl per side over 3 min. The injection needle was left in place for 2 min after injection to allow complete diffusion of the solution. Anti-BDNF antibody (Millipore, Billerica, MA, USA; cat# AB1513P) or control IgG (Sigma-Aldrich, cat# I5131) was dissolved in $1 \times$ PBS, and 0.5 μg was injected per side. The same dose of anti-BDNF blocks the formation of latent infantile memory at PN17 and the training-induced NMDA receptor Glu2B-GluN2A expression switch[22]. Recombinant BDNF (PeproTech, Rocky Hill, NJ, USA; cat# 450-02) was dissolved in $1 \times$ PBS and injected at 0.25 μg per side. This dose closes the infantile amnesia period and induces functional competence in rats trained at PN17. In all, 2 nmol of antisense oligodeoxynucleotides (ODNs) or relative scrambled sequences (SCR-ODNs) were injected per brain hemisphere. Sequences were as follows: PSD-95 antisense (AS), 5′-TGTGATCTCCTCATACTC-3′; PSD-95 SCR 5′-AAGCCCTTGTTCCCATTT-3′; c-Fos AS 5′-GCGTTGAAACCCGAGAACATC-3′; c-Fos SCR 5′-ACAAGAGCAT ACCGTGGTCCA-3′. The respective SCR-ODNs, which served as controls, contained the same base composition but in a random order and had no homology to sequences in GenBank. All ODNs were phosphorothioated on the three terminal bases of both the 5′ and 3′ ends to increase stability. ODNs were synthesized and reverse-phase cartridge purified by Gene Link (Hawthorne, NY, USA). To verify proper placement of the cannula implants, rats were euthanized at the end of the behavioral experiments, and their brains were frozen in isopentane, sliced in 40-μm coronal sections in a −20 °C cryostat, and examined under a light microscope for cannula placement. Rats with incorrect placement (5%) were discarded from the study.

**Western blot analysis.** Rats were euthanized, and their brains were rapidly removed and frozen in isopentane. DHC punches were obtained with a neuro punch (19 gauge; Fine Science Tools, Foster City, CA) from frozen brains mounted on a cryostat. Samples were homogenized in ice cold RIPA buffer (50 mM Tris base, 150 mM NaCl, 0.1% SDS, 0.5% Na-deoxycholate, 1% NP-40) with protease and phosphatase inhibitors (0.5 mM PMSF, 2 mM DTT, 1 mM EGTA, 2 mM NaF, 1 μM microcystin, 1 mM benzamidine, 1 mM sodium orthovanadate, and commercial protease and phosphatase inhibitor cocktails (Sigma-Aldrich)). Protein concentrations were determined using the Bio-Rad protein assay (Bio-Rad Laboratories, Hercules, CA, USA). Equal amounts of the total protein (20 μg per lane) were resolved on denaturing SDS-PAGE gels and transferred to the

Immobilon-FL Transfer membrane (Millipore) by electroblotting. Membranes were dried, reactivated in methanol, washed with water, and then blocked in TBS containing 5% (wt) milk for 1 h at room temperature. Membranes were then incubated with primary antibody overnight at 4 °C in the buffer recommended by the manufacturer. The following antibodies were used at the indicated dilutions: anti-Arc (1:10000, Synaptic System, Gottingen, Germany; cat# 156 003), anti-c-Fos (1:200, Millipore; cat# PC05), anti-Zif268 (1:1000, Cell Signaling Technology, Danvers, MA, USA; cat#4153 s), anti-Synaptophysin (1:1000, Cell Signaling Technology, cat# 5467), anti-PSD-95 (1:1000, Cell Signaling Technology, cat# 2507 s), anti-pAMPA Receptor GluA1 (Ser-845) (1:1000, Cell Signaling Technology, cat# 8084 s), anti-pAMPA Receptor GluA1 (Ser-831) (1:1000, Abcam, Cambridge, MA, USA; cat# ab109464). The membranes were then washed with TBS containing 0.2% Tween-20 (TBST) and incubated with species-appropriate fluorescently conjugated secondary antibody (goat anti-mouse IRDye 680LT (1:10,000) or goat anti-rabbit IRDye 800CW (1:10,000) from LI-COR Bioscience (Lincoln, NE, USA)) for 1 h at room temperature. Membranes were again washed in TBST and scanned using the Odyssey Infrared Imaging system (Li-Cor Bioscience). Data were quantified using pixel intensities with the Odyssey software (Li-Cor) according to the protocols of the manufacturer. Actin (1:20,000, Santa Cruz Biotechnology, Dallas, TX, USA; cat# sc-47778) was used as a loading control for all blots.

**Brain slice preparation.** Coronal brain slices (400 μm) were taken through dorsal regions of the rostral hippocampus. Slices were transferred to a chamber containing oxygenated artificial cerebrospinal fluid (ACSF) at 32 °C (30–45 min). Slices remained at room temperature for at least 1 h before recordings were made at 32 °C in oxygenated ACSF. ACSF contained (in mM): 125 NaCl, 4 KCl, 1.2 KH$_2$PO$_4$, 1.3 MgSO$_4$, 26 NaHCO$_3$, 15 glucose, 2.4 CaCl$_2$, and 0.4 L-ascorbic acid (pH = 7.4 when bubbled with 95% O$_2$/5% CO$_2$).

**Brain slice recordings.** Whole-cell current clamp recordings were carried out on CA1 pyramidal cells. The internal recording solution contained the following (in mM): 5 KCl, 127.5 K-gluconate, 10 HEPES, 2 MgCl$_2$, 0.6 EGTA, 2 ATP, 0.3 GTP, and 5 phosphocreatine (pH brought to 7.2 with KOH). The tip resistance of the patch electrode filled with internal solution was 5–10 MΩ. Access resistance was 15–30 MΩ, and was compensated by ~70%. Passive membrane and intrinsic firing properties were evaluated on the basis of responses to positive and negative current pulses (1500 ms). To determine AP threshold, incremental steps of current (20pA steps) were delivered at 0.2 Hz until a spike was evoked. Membrane resistance was calculated by passing −30 pA of current into the cell. All cells used for data analysis had at least a −50 mV resting potential and overshooting action potentials (APs). To investigate frequency–current curves (maximum firing rate), cells received depolarizing current injections from 20 to 200 pA (20 pA steps). AMPA receptor synaptic properties were obtained by stimulating Schaffer Collaterals with incremental current steps, while the cell was held at −80 mV in the presence of NMDA receptor antagonist D-2-amino-5-phosphonopentanoate (D-AP5, 50 μM—Tocris Bioscience, Bristol, UK). The peak EPSP amplitude was obtained in response to the maximum stimulus level just below the AP threshold. The difference between membrane voltage baseline and peak amplitude was used to measure max amplitude (mV). The amount of time between peak amplitude and return to baseline was used to measure AMPA receptor potential decay (ms). To measure paired-pulse responses, cells received paired pulses just below the AP threshold at 50 ms and 200 ms ISIs. The ratio of the second response to the first was used to determine if presynaptic release showed facilitation ($\geq 1.0$) or depression ($\leq 1.0$).

All procedures complied with the US National Institute of Health Guide for the Care and Use of Laboratory Animals and were approved by the New York University Animal Care Committees.

**Quantification and statistics.** Data were analyzed with the Prism 7 (GraphPad Software Inc.). No statistical methods were used to predetermine sample sizes, but our sample sizes are similar to those generally employed in the field. No randomization was used to collect the data. Statistical analyses were designed using the assumption of normal distribution and similar variance among groups, but this was not formally tested. The data were analyzed by one- or two-way analysis of variance (ANOVA) followed by post hoc tests. One-way ANOVA followed by Tukey's post hoc test was performed when comparing the groups for which a pairwise post hoc analysis of each group was required. One-way ANOVA followed by Dunnett's post hoc test was used when each group was compared with a single control group. Two-way ANOVA followed by Bonferroni post hoc tests was used when two factors were compared (e.g., treatment and testing). For F–I curves, a two-way mixed-model ANOVA (linear regression analysis w repeated measures) was used to verify a main effect of training on evoked firing rate during incremental current injection steps (0–200 pA). Intrinsic properties and AMPA receptor potential properties were compared using Tukey's honestly significant difference test, which is a post hoc analysis that accounts for between-group variance. For paired comparisons, Student's $t$ test was used. In all the experiments, both PN17 and PN24 females and males were included and analyzed as a single group, because statistical analyses of separate sex groups ($n = 2$–6) yielded no significant difference

(unpaired two-tailed Student's *t* test, *P* > 0.05). All analyses were two-tailed. The results were considered significant at *P* < 0.05.

**Reporting summary**. Further information on research design is available in the Nature Research Reporting Summary linked to this article.

## Data availability

The source data underlying Figs. 1–8 and Supplementary Figs. 1–3 are provided as a Source Data file.

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

## Acknowledgements
We thank Prof. Dan Sanes (Center for Neural Science, NYU) for helpful discussion and feedback. This work was supported by the DANA Foundation and R37 MH065635 to C.M.A.

## Author contributions
B.B., A.T., T.M.M., and C.M.A. contributed to designing experiments, analyzing the data, and writing the paper. B.B., A.T., and X.Z. performed the behavioral and molecular experiments; T.M.M. performed the electrophysiological experiments.

## Competing interests
The authors declare no competing interests.
