## [Peer Review File · Nature Communications]

Reviewers' comments:

Reviewer #1 (Remarks to the Author):

This is excellent work showing that infant rodents use mechanisms for memory formation that are distinct from adolescents. Further, the authors establish that spaced training generates memory competence that is domain specific. In my opinion, this work is of great general interest; it not only characterises memory formation in infant age, it also opens a novel window to investigate neurodevelopmental disorders. I think the manuscript could be strengthened by addressing the following questions:

1) c-Fos expression was shown to be upregulated 24 hours after training in PN17 rats. This is surprising because the up-regulation does not follow the profile of an 'immediate-early' gene. However, the authors used a c-fos promoter to inducibly activate neurons around the time of training (Figure 5). How can this work, if c-fos is not expressed around the time of training? Could it be that the transgenic system is leaky and that CNO some random cells are activated? Could one visualize which cells were activated? It would be good to discuss these issues in greater detail.

2) When considering the expression of PSD95, synaptophysin and phospho-forms of glutamate receptor subunits, the authors declare them as markers of synaptic maturation. I think this is not necessarily correct. These marker may be upregulated when synapses become bigger, or alternatively after synaptogenesis. Have the authors studied synapse numbers and synapse morphology? If not, it would be good to discuss these issues in greater detail.

3) There is a 24 hour delay between BDNF signaling and PSD95 upregulation. It would be good to discuss why there is such a long delay.

Reviewer #2 (Remarks to the Author):

This is an exciting paper that uses cutting-edge viral, transgenic, and clever behavioral techniques to tease out the mechanisms underlying learning and memory retrieval across multiple timepoints, and of "functional competence" i.e. the capacity for a brain to be primed to recall dormant memories. The study is elegant in its design and presents a clear story. More specifically, the authors examined if episodic learning in infants leads to time-specific changes in the hippocampus that are not present at subsequent stages of juvenile / adulthood. They show that learning in infancy leads to differential changes of the hippocampus at the molecular level compared to subsequent stages in life, and they go on to probe the contributions of AMPA receptors, BDNF, and PSD-95, as well as their behavioral relevance within the hippocampus domain of learning. Moreover, while this report provides a novel framework for interrogating the phenomenon of infantile amnesia in general, I have a few major and minor points that, if addressed (or some least discussed), will elevate this manuscript further into a well-controlled, meticulous, and systematic dissection of modulating memories across multiple timescales, which collectively would make the manuscript very suitable for publication in Nature Communications. I am delighted to sign off on this review as Steve Ramirez (Assistant professor of psychological and brain sciences at Boston University) and fully support the current manuscript's impact and integrity.

Major points:

1) On Page 5, how do the authors explain that IEGs show a specific time course early in life that seems to be different and longer lasting than in adulthood? A more fleshed out discussion on putative mechanisms of IEG upregulation in infancy vs adulthood would balance the discussion further, especially given that Fos / Arc / ZIP are very different when it comes to their functional roles within (and across) cells.

- 2) IEG expression was robust 48 hours later in fig 1, though the second IA training occurs 24 and not 48 hours apart, i.e. between P17 and P18 vs P17 and P19? Does this result hold true at the 48 hour timepoint?
- 3) While the ability to retrieve a memory from infancy seems to appear 1 and 6 days later from learning, do these memories persist at timepoints when memories are thought to be remote? While these particular experiments aren't necessary, I believe a discussion of whether or not infantile memories can "systems consolidate" in a manner consistent with episodic memory in adulthood would provide fruitful points of experiments for subsequent studies.
- 4) In Fig 5, is it possible to stimulate an ensemble of similar size not related to the precise learning experience—i.e. is just stimulating a cell ensemble in general or the discrete experience itself that sufficient to yield these behavioral effects?
- 5) What do the authors speculate the administration of systemic CNO is doing to the tagged ensemble vs. is CNO was administered to the hippocampus? As these perturbations can produce fundamentally different effects on local circuits and across systems, as well as their contributions to memory, (e.g. Garner et al. Science 2012 compared to Ramirez et al. Science 2013), it warrants a discussion on how these manipulations can influence behavior as well.

Minor Points

- 1) The discussion is very well written, and similar to Major Point 5, I believe that a nuanced paragraph on the advantages and limitations of double transgenic animals (since they can induce forebrain-wide labeling) vs local circuit perturbations can nicely balance the discussion by speculating on key networks or key nodes involved on producing these behavioral effects.
- 2) Post CNO, when was freezing measured?
- 3) In Garner et al., Science 2012, the researchers reported in a supplementary section that CNO administration sometimes led to local seizures, which sometimes can manifest as apparent immobility. While I appreciate that this is incredibly difficult to tease apart and surely do not expect the authors to systematically test as it is outside of this manuscript's scope, do they have any way of ruling out this potential confound?
- 4) Page 3—please define BDNF on its first use
- 5) Page 6—the phrasing can be changed to "training also significantly increased"
- 6) For the transgenic animals, were male and/or females utilized?
- 7) For the behavioral tests, how was a cap of 900 seconds determined?
- 8) Page 27: the placement of the "novel object" can be reworded to "object in novel location" to prevent the reader for misinterpreting the behavior as novel object recognition.

Overall, I applaud the authors on this manuscript, as it is a thoughtful, provocative, and highly novel contribution to the field.

Reviewer #3 (Remarks to the Author):

This manuscript by Travaglia and colleagues follows up on a recently published paper by the same group (Travaglia et al., 2017) demonstrating that early life episodic memories are not lost but remain stored long term. The current study extends those findings by demonstrating that early life experiences can selectively mature learning and memory abilities. Overall, the experiments appear technically sound and are clearly presented. The data is solid and well analyzed. The paper is compact and well written. However, there are some concerns that should be addressed in order to strengthen the conclusion of the study.

Major comments:

1. The findings of experience-dependent selective maturations are extremely interesting. The authors claim that manipulating hippocampal cell ensembles marked by activation upon learning is sufficient to produce an experience-specific maturation of functional competence (Fig. 5g). The concern rests in whether the CFC and the nOL activate distinct populations of hippocampal cell ensembles or network during learning. If so, behavioral experiences that share an activated population of neurons during training affect the experience-dependent selective maturations. In addition, did silencing of ensembles corresponding to the CFC suppress the maturation of functional competences?

2. The authors state that PSD95 is essential for infantile, but not juvenile, memory storage. The authors should consider or speculate what existing factors contribute to this developmental switch in the synaptic mechanisms of memory formation. In addition, it is unclear how PSD95 contributes learning-induced functional maturation of the hippocampal memory system. Because PSD95 is important in coupling the NMDA receptor to pathways that control bidirectional synaptic plasticity and learning, the authors should determine whether the developmental switches in NMDA receptor subunit expression correlates with early emergence of adult-like memory retention in infants.

3. The authors show that training at PN17, not PN24, caused a significant increase in the amplitude of evoked EPSP and a significant decrease in EPSP decay time. However, it is unclear how this change is caused. Is AMPA receptor function changed, or the subtype of AMPA receptor activation altered? Is the recording in the presence of NMDA receptor antagonists? If not, it could not be presented as AMPA amplitude and decay. It will be also supported by electrophysiological data such as mEPSC and AMPA/NMDA ratio measurements.

4. The authors claim that the infant animals respond with a slow and long-lasting IEG inductions following learning compared to adults. However, there is little discussion about the possible reasons responsible for this finding. Some attempt to discuss this finding should be made.

5. It is conceivable that BDNF is released during learning training. However, the authors fail to correlate the time profiles of BDNF release and subsequent increase in PSD95 expression.

6. Could the authors observe the inhibitory effect of PSD95 AS-ODN in the EPSP experiments following learning?

Minor comments:

1. Fig. 1a, the internal standard for pGluA1 would be GluA1 better than actin.

2. Were the EPSPs recorded in the presence of GABA_A receptor antagonist?

Responses to reviewers

We thank the Reviewers for their valuable feedback, comments, and suggestions, which helped us to clarify and strengthen our manuscript. We have substantially revised our manuscript and significantly expanded our results and the discussion to address important points raised by the Reviewers. Please find below our point-by-point response to the reviewers' comments. Changes in our manuscript are identified by **red** font.

Also, given that Benjamin Bessières conducted all the experiments of the revision, we have switched the order of the co-first authors (Bessières and Travaglia). Nevertheless, they both remain co-first authors.

Reviewers' comments:

Reviewer #1 (Remarks to the Author):

This is excellent work showing that infant rodents use mechanisms for memory formation that are distinct from adolescents. Further, the authors establish that spaced training generates memory competence that is domain specific. In my opinion, this work is of great general interest; it not only characterizes memory formation in infant age, it also opens a novel window to investigate neurodevelopmental disorders. I think the manuscript could be strengthened by addressing the following questions:

1) c-Fos expression was shown to be upregulated 24 hours after training in PN17 rats. This is surprising because the up-regulation does not follow the profile of an 'immediate-early' gene. However, the authors used a c-fos promoter to inducible activate neurons around the time of training (Figure 5). How can this work, if c-fos is not expressed around the time of training? Could it be that the transgenic system is leaky and that CNO some random cells are activated? Could one visualize which cells were activated? It would be good to discuss these issues in greater detail.

Response:

The Reviewer raises an interesting question regarding the unusual long-lasting increase of immediate early genes (IEGs) following inhibitory avoidance (IA) training at postnatal day 17 (PN17), compared to PN24 or adult age. Specifically, training at PN17 increases IEGs for more than 24 hours (h). The Reviewer asks how the cfos-htTA/tetO-hM3Dq mice can be used to induce the expression of hM3Dq in response to training if c-Fos is not induced around the time of training.

Concerning this last point, it should be reminded that pre-weaning mice received doxycycline (DOX) by lactation from the mothers, which were fed with DOX immediately after the pups underwent contextual fear conditioning (CFC), and then remained on DOX in order to prevent non-learning related expression of the hM3Dq receptors (see Methods section). This DOX delivery via lactation is likely to start blocking the expression of new hM3Dq receptors with a small delay. Therefore, this delay may be sufficient to allow for some c-Fos, hence hM3Dq expression. In agreement with this explanation, the delivery of the ligand CNO, seven days after training, resulted in long-lasting memory expression (new Fig. 7c).

Nevertheless, to fully address the Reviewer's question, we performed a new experiment in the *cfos-htTA/tetO-hM3Dq* mice. We employed CFC training at PN17 and determined the time course (30 min, 24h and 48h) of c-Fos induction in the dorsal hippocampus (dHC) using western blot analyses. Untrained controls were euthanized at matched timepoints. As shown in the new Fig. 7b, CFC training in mice, like IA in rats, induces a gradual and long-lasting upregulation of c-Fos in the dHC, which peaks 24 h after training and returns to baseline one day later. Notably, at 30 min after training (although lower than the 24-hour time-point), the level of c-Fos induced by CFC is significantly increased compared to control level (unpaired t-test; $t=3.236$; $df=8$; $p=0.0119$). These data indicate that c-Fos induction in mouse starts rapidly after CFC training.

Moreover, we added additional discussion points regarding the slow and sustained kinetics of IEG induction following training at PN17 compared to later ages. Specifically, we added the following text (page 18):

“These distinctive molecular regulations of the infant hippocampus suggest that the hippocampus-dependent memory system at this age, compared to later developmental stages, is biologically different. The bases for these unique biological regulations remains to be understood, and, at present, we can only make some speculations. For example, the gradual and sustained upregulation of IEGs following training at PN17 may reflect a higher level of hippocampal synaptic plasticity and neuronal excitability at this age compared to later ages (Harris and Teyler 1984; Dumas 2005; Nabel and Morishita 2013; Marin-Burgin et al. 2012). This hypothesis is in line with the fact that basal expression of IEGs as well as of molecular markers of synaptic plasticity and neuronal excitability, such as cAMP response element-binding protein (CREB) (Josselyn and Frankland 2018) and NMDA receptor GluN2B subunit (Yashiro and Philpot 2008; Lee et al. 2010; Gambrill and Barria 2011) is higher in the dorsal hippocampus of infant rats compared to adults (Travaglia et al., 2016a). Moreover, the prolonged IEG induction in the infant brain could result from a lower GABAergic inhibition (Travaglia et al. 2016a; Donato et al., 2017), as suggested by greater IEG expression in the lateral amygdala following chemogenetic inhibition of parvalbumin-expressing interneurons during memory formation in adult mice (Morrison et al. 2016).”

2) When considering the expression of PSD95, synaptophysin and phospho-forms of glutamate receptor subunits, the authors declare them as markers of synaptic maturation. I think this is not necessarily correct. These marker may be upregulated when synapses become bigger, or alternatively after synaptogenesis. Have the authors studied synapse numbers and synapse morphology? If not, it would be good to discuss these issues in greater detail.

Response:

We did not quantify changes in synapse number and morphology following IA learning in infant rats. Thus, the Reviewer is correct that we cannot exclude the possibility that IA training at PN17 induces the formation of new synapses in the dHC. We therefore agree that both synapse maturation as well as synaptogenesis may occur with training, and we corrected the language in our revised manuscript.

3) There is a 24-hour delay between BDNF signaling and PSD95 upregulation. It would be good to discuss why there is such a long delay.

Response:

We blocked BDNF by injecting anti-BDNF blocking antibody in the dHC before training and then determined whether this treatment affects the learning-dependent increase in hippocampal PSD-95 (Fig. 2a). Learning at PN17 leads to an increase in PSD-95 that follows a relatively slow kinetics, peaking at 24 hours after training, and remaining elevated at 48 hours (Fig. 1b). Our experiment tested whether blocking BDNF around the time of training affects the PSD95 upregulation measured 24 hours later. We found that to be the case.

Although the mechanisms by which BDNF regulation around training controls the expression of PSD-95 are not known, our data clearly show that BDNF is upstream the PSD-95 upregulation.

Reviewer #2 (Remarks to the Author):

This is an exciting paper that uses cutting-edge viral, transgenic, and clever behavioral techniques to tease out the mechanisms underlying learning and memory retrieval across multiple time points, and of “functional competence” i.e. the capacity for a brain to be primed to recall dormant memories. The study is elegant in its design and presents a clear story. More specifically, the authors examined if episodic learning in infants leads to time-specific changes in the hippocampus that are not present at subsequent stages of juvenile / adulthood. They show that learning in infancy leads to differential changes of the hippocampus at the molecular level compared to subsequent stages in life, and they go on to probe the contributions of AMPA receptors, BDNF, and PSD-95, as well as their behavioral relevance within the hippocampus domain of learning. Moreover, while this report provides a novel framework for interrogating the phenomenon of infantile amnesia in general, I have a few major and minor points that, if addressed (or some least discussed), will elevate this manuscript further into a well-controlled, meticulous, and systematic dissection of modulating memories across multiple timescales, which collectively would make the manuscript very suitable for publication in Nature Communications. I am delighted to sign off on this review as Steve Ramirez (Assistant professor of psychological and brain sciences at Boston University) and fully support the current manuscript’s impact and integrity.

Major points:

1) On Page 5, how do the authors explain that IEGs show a specific time course early in life that seems to be different and longer lasting than in adulthood? A more fleshed out discussion on putative mechanisms of IEG upregulation in infancy vs adulthood would balance the discussion further, especially given that Fos / Arc / ZIF are very different when it comes to their functional roles within (and across) cells.

Response:

The Reviewer raised a very interesting point on the sustained upregulation of the IEGs by learning at PN17. This point was also raised by Reviewer 1. We

agree and expanded the Discussion on those findings (page 18). Specifically we added the following:

“These distinctive molecular regulations of the infant hippocampus suggest that the hippocampus-dependent memory system at this age, compared to later developmental stages, is biologically different. The bases for these unique biological regulations remains to be understood, and, at present, we can only make some speculations. For example, the gradual and sustained upregulation of IEGs following training at PN17 may reflect a higher level of hippocampal synaptic plasticity and neuronal excitability at this age compared to later ages (Harris and Teyler 1984; Dumas 2005; Nabel and Morishita 2013; Marin-Burgin et al. 2012). This hypothesis is in line with the fact that basal expression of IEGs as well as of molecular markers of synaptic plasticity and neuronal excitability, such as cAMP response element-binding protein (CREB) (Josselyn and Frankland 2018) and NMDA receptor GluN2B subunit (Yashiro and Philpot 2008; Lee et al. 2010; Gambrill and Barria 2011) is higher in the dorsal hippocampus of infant rats compared to adults (Travaglia et al., 2016a). Moreover, the prolonged IEG induction in the infant brain could result from a lower GABAergic inhibition (Travaglia et al. 2016a; Donato et al., 2017), as suggested by greater IEG expression in the lateral amygdala following chemogenetic inhibition of parvalbumin-expressing interneurons during memory formation in adult mice (Morrison et al. 2016).”

2) IEG expression was robust 48 hours later in fig 1, though the second IA training occurs 24 and not 48 hours apart, i.e. between P17 and P18 vs P17 and P19? Does this result hold true at the 48 hour timepoint?

Response:

The second IA training was given 24 hours after the first one because 24 hours is when all the molecular changes investigated peaked (Fig. 1). Therefore, we reasoned that it was an appropriate time point to be functionally tested. Indeed, a second trial given 24 h after the first one led to memory competence. The Reviewer here asks an interesting question: given that most changes persists at 48 hours after training, would a second learning trial given 48 hours after the first training be able to still induce functional competence?

We performed the suggested experiment and reported the new data in the new Fig. 4b: rats trained in IA at PN17, and then again 48 h later (PN17+PN19); two other groups of rats underwent a single training session, either at PN17 or at PN19. (Fig. 4b). The rats that received IA training at PN17 and PN19 showed a robust and long-lasting IA memory at both 1 and 5 days after the second training. As expected, the rats trained at PN17 or at PN19 showed a rapid forgetting 1 day after training, which persisted at PN24. Hence, indeed, the results hold true at 48 hours: a second training trial given 48 hours after the first training (at PN17) promotes memory competence.

3) While the ability to retrieve a memory from infancy seems to appear 1 and 6 days later from learning, do these memories persist at timepoints when memories are thought

to be remote? While these particular experiments aren't necessary, I believe a discussion of whether or not infantile memories can "systems consolidate" in a manner consistent with episodic memory in adulthood would provide fruitful points of experiments for subsequent studies.

Response:

We believe that the Reviewer is asking whether memory competence induced by 2 training trials one at PN17 and the other PN18, lasts for weeks, which is typical of remote memories that have undergone system consolidation.

We carried out this experiment, which is now depicted in the new Fig. 4d. We trained rats in IA at PN17 and PN18, then we tested their memory 29 days later (at PN47). As shown in Fig. 4d, the rats exhibited a significant memory retention at this remote time point. This memory, however, was weaker compared to the memory measured at 1 or 6 days after training, presumably due to some physiological forgetting over this long temporal window.

Thus, it seems that infantile memories can "systems consolidate" in a manner consistent with episodic memory in adulthood, although we cannot exclude that the system consolidation process in an infant brain may be different. These questions shall be investigated in future studies. An additional proof that infantile memories are very long lasting was also shown in our previous study (Travaglia et al. 2016b): learning at PN17 created a memory that although not expressed, was stored for at least 4 weeks. In fact, a re-instatement protocol given 4 weeks after a single training at PN17 resulted in strong and sustained memory expression (Travaglia et al. 2016b).

Another observation in line with the standard model of systems-level memory consolidation is that a bilateral injection of the neural activity blocker GABA_A agonist muscimol in the dorsal hippocampus (dHC) blocked the formation of IA memory at PN17, but not memory reinstatement 7-9 days after training (Travaglia et al. 2016b). Although, the contribution of cortical areas in infantile memory consolidation remains to be explored, our data suggest that infantile memories undergo a systems-level reorganization. We have added a point of discussion on page 21.

4) In Fig. 5, is it possible to stimulate an ensemble of similar size not related to the precise learning experience—i.e. is just stimulating a cell ensemble in general or the discrete experience itself that sufficient to yield these behavioral effects?

Response:

The Reviewer raises another interesting question, that is, the specificity of the cell ensemble activated by training. We believe that the results presented in the new Fig. 7h (old Fig. 5g) address this point.

Artificial stimulation by CNO of the cell ensembles activated by CFC learning produced an experience-specific maturation of functional competence.

In fact, artificial reactivation (by CNO) of the cell ensemble activated earlier by CFC learning, while undergoing a different learning experience - novel object location (nOL), promoted functional competence only for CFC and not for nOL memory. These results showed that the stimulation of a cell ensemble not related

to a specific learning experience is not sufficient to promote functional competence for that experience.

Hence, our data suggest that a learning experience in infancy activates a specific cellular ensemble that supports selective functional maturation of that learning domain, but not of distinct types of hippocampal learning domains.

5) What do the authors speculate the administration of systemic CNO is doing to the tagged ensemble vs. is CNO was administered to the hippocampus? As these perturbations can produce fundamentally different effects on local circuits and across systems, as well as their contributions to memory, (e.g. Garner et al. Science 2012 compared to Ramirez et al. Science 2013), it warrants a discussion on how these manipulations can influence behavior as well.

Response:

We thank the Reviewer for raising this important point. The reason why we opted for intraperitoneal (i.p.) CNO administration is because cannulae implantation in infant mice is very challenging given the very small size of the mouse brain at this age. The systemic vs. local effects of CNO treatments are debated. We added a Discussion point to highlight this issue. Specifically, we included the following text (page 21):

“In our experiments employing chemogenetic stimulation, we used intraperitoneal CNO administration because cannulae implantation in infant mice is very challenging due to the very small size of the mouse brain at this age. Therefore, because systemic administration of CNO to the cfos-htTA/tetO-hM3Dq mice triggers a forebrain-wide activation of the hM3Dq receptors (Garner et al. 2012), our data cannot exclude the possibility that, in addition to the hippocampus, other forebrain regions were activated and contributed to the acquisition of functional competence. Furthermore, the specificity of systemically administered CNO and/or its metabolites have been debated (for review Roth 2016). Therefore, to limit the potential side effects of systemic CNO injection, we used a low dose of CNO (i.e. 0.5 mg/kg), which had been previously found effective in activating hM3Dq receptors in the hippocampus (Alexander et al. 2009; Garner et al. 2012). Finally, to control for CNO-induced non-memory-related effects we injected the same dose of CNO to control mice that did not express the hM3Dq receptors.”

Minor Points:

1) The discussion is very well written, and similar to Major Point 5, I believe that a nuanced paragraph on the advantages and limitations of double transgenic animals (since they can induce forebrain-wide labeling) vs local circuit perturbations can nicely balance the discussion by speculating on key networks or key nodes involved on producing these behavioral effects.

Response:

We agree and, as mentioned in the response to point 5, we added a discussion point on this matter (page 21).

2) Post CNO, when was freezing measured?

Response:

Freezing was measured 30 min after CNO injection during the first test at PN24 and then again 7 days later at PN31 for the experiment depicted in the new Fig. 7c. In addition, for the experiment shown in the new Fig. 7h, freezing was measured 24 hours after CNO injection and then again 5 days later (Fig. 7h). We better clarified the schedule description in the text (pages 16 and 17).

3) In Garner et al., Science 2012, the researchers reported in a supplementary section that CNO administration sometimes led to local seizures, which sometimes can manifest as apparent immobility. While I appreciate that this is incredibly difficult to tease apart and surely do not expect the authors to systematically test as it is outside of this manuscript's scope, do they have any way of ruling out this potential confound?

Response:

Although Garner et al. (2012) reported that an i.p injection of CNO at 0.5 mg/kg resulted in behavioral signs of seizure activity in approximately 20% of mice during fear conditioning, we never observed any behavioral signs of seizure. Although mice may have had epilepsy without obvious seizure, we have not observed any sign of it. Moreover, our control mice, which did not express hM3Dq and were injected with the same dose of CNO, showed a very low level of freezing compared to the hM3Dq -expressing mice (new Fig. 7c,h), indicating that the freezing observed in our experiments is likely related to CFC memory expression rather than CNO-induced other effects.

4) Page 3—please define BDNF on its first use

Response: Corrected.

5) Page 7—the phrasing can be changed to “training also significantly increased”

Response: Corrected.

6) For the transgenic animals, were male and/or females utilized?

Response:

Both male and female transgenic mice were used for our experiments. Females and males were analyzed as a single group because preliminary statistical analyses of female and male groups (n = 2-6) showed no significant difference in value distribution (unpaired two-tailed Student's t-test, $P > 0.05$). Although these numbers are too low for any robust statistical analysis, we decided to group the subjects and not pursue sex-related questions. This information is detailed in the Methods section.

7) For the behavioral tests, how was a cap of 900 seconds determined?

Response:

The cap of 900 seconds in inhibitory avoidance latency has been chosen in order to have a large temporal window for recording the latencies of animals and avoid having ceiling effects on latencies.

8) Page 31: the placement of the “novel object” can be reworded to “object in novel location” to prevent the reader for misinterpreting the behavior as novel object recognition.

Response: Corrected (page 25 of our revised manuscript).

Overall, I applaud the authors on this manuscript, as it is a thoughtful, provocative, and highly novel contribution to the field.

Reviewer #3 (Remarks to the Author):

This manuscript by Travaglia and colleagues follows up on a recently published paper by the same group (Travaglia et al., 2016) demonstrating that early life episodic memories are not lost but remain stored long term. The current study extends those findings by demonstrating that early life experiences can selectively mature learning and memory abilities. Overall, the experiments appear technically sound and are clearly presented. The data is solid and well analyzed. The paper is compact and well written. However, there are some concerns that should be addressed in order to strengthen the conclusion of the study.

Major comments:

1. The findings of experience-dependent selective maturations are extremely interesting. The authors claim that manipulating hippocampal cell ensembles marked by activation upon learning is sufficient to produce an experience-specific maturation of functional competence (Fig. 5g). The concern rests in whether the CFC and the nOL activate distinct populations of hippocampal cell ensembles or network during learning. If so, behavioral experiences that share an activated population of neurons during training affect the experience-dependent selective maturations. In addition, did silencing of

ensembles corresponding to the CFC suppress the maturation of functional competences?

Response:

As mentioned by the Reviewer, our results show that artificial stimulation of the hippocampal cell ensembles activated with learning at PN17 (new Fig. 7h, old Fig. 5g) is sufficient to produce an experience-specific maturation of functional competence.

This conclusion implies that similar learning experiences (such as two IA or two CFC trainings, given in the same or different contexts) would activate overlapping populations of hippocampal cell ensembles, hence promoting selective functional maturation of that learning domain. Conversely, two different types of learning (such as IA + nOL, or CFC + nOL) would activate distinct populations of hippocampal cell ensembles, which would not allow the maturation of functional competences for these learning events. Several converging lines of evidence in rats and mice brought us to our conclusion:

1) In rats, whereas two IA in the same or different contexts, or two nOL with the same or different object pairs and spatial configurations produce functional competence (new Fig. 4 and new Fig. 5a-c), one IA and one nOL do not (new Fig. 5d,e). Thus, functional competence transfers to similar types of learning but not to distinct ones. We found similar results using CFC and nOL in mice (new Fig. 7e-g).

2) In mice, we also showed that the artificial stimulation of the cell ensembles activated upon one learning task (CFC) matures functional competence for this specific task but not for a different learning task (nOL) (new Fig. 7c,h).

3) We also found that the molecular changes elicited by the first learning are necessary to promote functional competence. In fact, inhibition of training-induced PSD95 upregulation following the first IA learning blocked the acquisition of functional competence following the second IA training (new Fig. 4e).

In addition, the Reviewer is asking whether "silencing" the CFC cell ensembles blocks the maturation of functional competence. To address this question, one possibility would be to use *cfos*-tTA/tetO-hM4Di double transgenic mice, which express the hM4Di protein under the regulation of the *c-fos* promoter in a doxycycline (Dox)-dependent manner. CNO administration in these mice would therefore silence the cell ensemble activated at training. Using this method, we could silence the cell ensemble activated by the first CFC learning at PN17 and test the effect on memory competence. However, this experiment would necessitate to establish a new mouse colony, which would require at least 6-8 months. We therefore opted to address this question by targeting the endogenous activation of *c-Fos* following learning (since *c-Fos* promoter activation is used in the *cfos*-htTA/tetO-hM3Dq mice).

We employed antisense oligodeoxynucleotides (AS-ODN) to disrupt the induction of *c-Fos* evoked by IA learning at PN17 in rats and tested whether this inhibition affected IA memory functional competence following a second IA trial at PN18. As depicted in the new Fig. 6, *c-Fos* AS-ODN injected after the first training completely blocked the learning-dependent increase in *c-Fos* at 24h (Fig. 6a). Notably, *c-Fos* AS-ODN prevented the acquisition of functional competence (Fig. 6b). These data indicate that the *c-Fos*-dependent activation of the cell ensemble

engaged upon the first learning experience is necessary for the maturation of functional competence.

2. The authors state that PSD95 is essential for infantile, but not juvenile, memory storage. The authors should consider or speculate what existing factors contribute to this developmental switch in the synaptic mechanisms of memory formation. In addition, it is unclear how PSD95 contributes learning-induced functional maturation of the hippocampal memory system. Because PSD95 is important in coupling the NMDA receptor to pathways that control bidirectional synaptic plasticity and learning, the authors should determine whether the developmental switches in NMDA receptor subunit expression correlates with early emergence of adult-like memory retention in infants.

Response:

The Reviewer raises an interesting point. We have added some discussion on the potential factors that may contribute to the developmental switch in the synaptic mechanisms of memory formation. Specifically, we added the following text (page 19):

“The differential biological changes in markers of synapse formation/maturation found in the hippocampus at PN17 is in agreement with and expands on our previous findings (Travaglia et al. 2016b, 2018) showing that learning in infancy engages mechanisms typically recruited during critical periods (Hensch 2005). Collectively, these data indicate that the infant hippocampus recruits differential synaptic mechanisms for the formation of infantile memories and further support our hypothesis of the existence of a developmental temporal window during which maturation of learning and memory abilities and its underlying biology occur (Alberini and Travaglia 2017).”

Regarding the PSD95 contribution to learning-induced functional maturation of the hippocampal memory system, our results suggest that one mechanism by which PSD95 upregulation at PN17 upon training might contribute to the functional maturation of the hippocampal memory system would be by allowing the stability of the AMPA receptors in excitatory synapses. Indeed, previous *in vitro* and *ex vivo* studies reported that PSD95 plays critical roles in synapse formation and maturation by interacting with, trafficking and stabilizing AMPA receptors in the postsynaptic membrane (Chen et al. 2000; Nicoll et al. 2006; Schnell et al. 2002). Knockdown of PSD-95 prevents spine stabilization after LTP induction (Ehrlich et al., 2007) and leads to silent synapse formation reflected by a decrease in AMPA receptor and an increase in GluN2B-containing NMDA receptor expression in the synapse (Beique et al., 2006; Huang et al., 2015). In line with these previous observations, we found that PSD95 at PN17 is required for learning-induced increase in evoked AMPA receptor EPSPs amplitude and decrease in AMPA decay times (new Fig. 3).

Finally, regarding “whether the developmental switches in NMDA receptor subunit expression correlates with early emergence of adult-like memory retention in infants”: we have shown in previous studies (Travaglia et al. 2016b, 2018) that learning in PN17 (infant), but not in PN24 (juvenile) rats when memory is

expressed long-term, induces a BDNF- and mGluR5-dependent increase in the ratio of NMDA receptor subunits GluN2A/GluN2B, and a functional switch toward synaptic GluN2A engagement. We showed that activation of mGluR5 or BDNF administration in the dorsal hippocampus immediately after IA training at PN17 induces the GluN2B/GluN2A switch and promotes the emergence of adult-like memory in infant rats (Travaglia et al. 2016b).

3. The authors show that training at PN17, but not PN24, caused a significant increase in the amplitude of evoked EPSP and a significant decrease in EPSP decay time. However, it is unclear how this change is caused. Is AMPA receptor function changed, or the subtype of AMPA receptor activation altered? Is the recording in the presence of NMDA receptor antagonists? If not, it could not be presented as AMPA amplitude and decay. It will be also supported by electrophysiological data such as mEPSC and AMPA/NMDA ratio measurements.

Response:

The Reviewer raises another interesting question. The mechanisms by which training at PN17 but not at PN24 leads to a significant increase in the amplitude of evoked EPSP and a significant decrease in EPSP decay time are still unknown. As shown by our multiple sets of data in this study and in our previous ones (Travaglia et al. 2016a,b, 2018), the biological composition of the hippocampus at PN17 is significantly different than that at PN24. However, we have added speculations (page 18) for why these differences are observed. We suggest that:

“Also unknown are the mechanisms by which learning at PN17 but not at PN24 leads to PSD-95 dependent changes in AMPA receptor responses in the hippocampus. Possibly, the two ages may have different rates or regulations of receptor trafficking, of synaptic levels of AMPA receptor subunits (GluA1-4), and/or different learning-induced post-translational modifications. Based on our findings, we speculate that infantile learning, via increased phosphorylation of GluA1 subunits at Ser-831 and Ser-845 may lead to changes in the AMPA receptor biophysical properties and synaptic expression, which might result in an increase in AMPA receptor EPSP amplitude and decrease in decay times. Finally, it cannot be excluded that learning at PN17 involves other types of changes to AMPA receptors, which still remain to be identified.”

In agreement with our hypothesis, previous studies had shown that phosphorylation of the C-terminal tail of GluA1 at Ser-831 leads to an enhanced single channel conductance (Mammen et al. 1997; Derkach et al. 1999). Moreover, phosphorylation of GluA1 at Ser-845, increases the availability of GluA1-containing AMPA receptors at extrasynaptic pools for synaptic insertion (Esteban et al. 2003; Dias et al. 2012) and increases the channel opening probability of the AMPA receptor and the current peak (Roche et al. 1996; Banke et al. 2000).

Also, we apologize for the missing information from the Methods, which has now been added (page 30). The recordings were performed in the presence of the NMDA receptor antagonist AP-5 (50 μ M), which was added to the ACSF bath. Cells were held at -80 mV in current clamp. AP-5 blocks the NMDA receptor

glutamate binding site. Holding the cell at -80 mV keeps the magnesium block into the NMDA channel and is past the chloride reversal potential at ~ -70 mV, effectively removing contemporaneously GABAergic signal from the trace. In fact, in this experiment, the addition of AP-5 is slightly redundant since clamping the cells at -80 mV already prevents the magnesium block removal from the NMDA pore.

Finally, to the Reviewer point that our conclusions could be supported by additional electrophysiological data such as mEPSC and AMPA/NMDA ratio measurements, we recorded these experiments in current clamp and not in voltage clamp where AMPA/NMDA ratio measurements are possible. We specifically decided to measure AMPA receptor potentials, as NMDA receptor data had been shown and discussed in the previous study (Travaglia et al. 2016b). Current clamp experiments also allowed us to measure whether cellular properties were altered by IA training (e.g. membrane potential, resistance, firing rates), which cannot be measured in voltage clamp.

4. The authors claim that the infant animals respond with a slow and long-lasting IEG inductions following learning compared to adults. However, there is little discussion about the possible reasons responsible this finding. Some attempt to discuss this finding should be made.

Response:

Please see response to Reviewer #2, point #1.

5. It is conceivable that BDNF released during learning. However, the authors fail to correlate the time profiles of BDNF release and subsequent increase in PSD95 expression.

Response:

Please see response to Reviewer #1, point #3.

6. Could the authors observe the inhibitory effect of PSD95 AS-ODN in the EPSP experiments following learning?

Response:

We thank the Reviewer for this suggestion. We carried out the suggested experiment, and, as shown in the new Fig. 3e-h, we found that the IA learning-induced changes in AMPA receptor responses is dependent on hippocampal PSD-95.

Specifically, PN17 rats received IA training, followed by two bilateral hippocampal injections of the PSD-95 AS-ODN or scrambled ODN that served as control (SCR-ODN). Whole-cell current clamp recordings from pyramidal neurons in the granule cell layer of area CA1 of the dHC were performed at PN18.

Recordings were also conducted in parallel on naïve rats at the same age injected with scrambled ODN.

For each cell, we first recorded active and passive membrane properties (Fig. 3e). We observed no effect of training or PSD-95 AS-ODN injections on firing rates (as a function of injected current), resting membrane potential or membrane resistance, suggesting that intrinsic excitability of CA1 hippocampal pyramidal cells did not change in response to IA training with PSD-95 AS-ODN-mediated knock down. We also observed no effect of training or PSD-95 AS-ODN on paired pulse ratios (Fig. 3h), suggesting that glutamatergic pre-synaptic release kinetics was not affected.

In contrast, the learning-dependent increase in evoked AMPA receptor EPSP amplitude and the decrease in EPSP decay times, observed 24h after PN17 training was abolished by the bilateral hippocampal injections of the PSD-95 AS-ODN (Fig. 3f,g).

Minor comments:

1. Fig. 1a, the internal standard for pGluA1 would be GluA1 better than actin.

Response:

Actin was used as internal standard for all our western blot experiments. Furthermore, no change was detected in the level of GluA1 following IA training at PN17 compared to untrained control (Supplementary Fig. 3).

2. Were the EPSPs recorded in the presence of GABA_A receptor antagonist?

Response:

No, the evoked AMPA receptor EPSPs were not recorded in the presence of GABA_A receptor antagonist. As indicated in the Method section of our revised manuscript, the recordings were done in ACSF with AP-5 (NMDAR antagonist). Cells were held at -80 mV in current clamp, hence past the chloride reversal -70 to -75 mV, which allows the blockage of GABA_A potential. We thought that GABA_A receptor antagonist would cause all the cells in the bath to become depolarized, leading to excitotoxicity, unless very low levels were used. Moreover, here we did not intent to show GABA_B receptor specificity, which would justify to use GABA_A receptor antagonist.

References

Alberini, C.M. & Travaglia, A. Infantile Amnesia: A Critical Period of Learning to Learn and Remember. *J. Neurosci.* **37**, 5783-5795 (2017).

Alexander, G.M., Rogan, S.C., Abbas, A.I., Armbruster, B.N., Pei, Y., Allen, J.A., Nonneman, R.J., Hartmann, J., Moy, S.S., Nicoletis, M.A., McNamara, J.O., Roth, B.L.

Remote control of neuronal activity in transgenic mice expressing evolved G protein-coupled receptors. *Neuron* **63**, 27-39 (2009).

Banke, T.G., Bowie, D., Lee, H., Huganir, R.L., Schousboe, A., Traynelis, S.F. Control of GluR1 AMPA receptor function by cAMP-dependent protein kinase. *J. Neurosci.* **20**, 89-102 (2000).

Béïque, J.C., Lin, D.T., Kang, M.G., Aizawa, H., Takamiya, K., Huganir, R.L. Synapse-specific regulation of AMPA receptor function by PSD-95. *Proc Natl Acad Sci U S A.* **103**, 19535-19540 (2006).

Chen, L., Chetkovich, D.M., Petralia, R.S., Sweeney, N.T., Kawasaki, Y., Wenthold, R.J., Brecht, D.S., Nicoll, R.A. Stargazin regulates synaptic targeting of AMPA receptors by two distinct mechanisms. *Nature* **408**, 936-943 (2000).

Derkach, V., Barria, A., Soderling, T.R. Ca²⁺/calmodulin-kinase II enhances channel conductance of alpha-amino-3-hydroxy-5-methyl-4-isoxazolepropionate type glutamate receptors. *Proc Natl Acad Sci U S A.* **96**, 3269-3274 (1999).

Dias, R.B., Ribeiro, J.A., Sebastião, A.M. Enhancement of AMPA currents and GluR1 membrane expression through PKA-coupled adenosine A(2A) receptors. *Hippocampus* **22**, 276-291 (2012).

Donato, F., Jacobsen, R.I., Moser, M.B., Moser, E.I. Stellate cells drive maturation of the entorhinal-hippocampal circuit. *Science* **355**, 6330 (2017).

Ehrlich, I., Klein, M., Rumpel, S., Malinow, R. PSD-95 is required for activity-driven synapse stabilization. *Proc Natl Acad Sci U S A* **104**, 4176-4181 (2007).

Esteban, J.A., Shi, S.H., Wilson, C., Nuriya, M., Huganir, R.L., Malinow, R. PKA phosphorylation of AMPA receptor subunits controls synaptic trafficking underlying plasticity. *Nat. Neurosci.* **6**, 136-143. (2003).

Gambrill, A.C, Barria, A. NMDA receptor subunit composition controls synaptogenesis and synapse stabilization. *Proc. Natl. Acad. Sci. USA* **108**, 5855–5860 (2011).

Garner, A.R. et al. Generation of a synthetic memory trace. *Science* **335**, 1513-1516 (2012).

Harris, K.M., Teyler, T.J., Developmental onset of long-term potentiation in the area CA1 of the rat hippocampus. *J. Physiol.* **346**, 27–48 (1984).

Hensch, T. K. Critical period plasticity in local cortical circuits. *Nat. Rev. Neurosci.* **6**, 877-888 (2005).

Huang, X., Stodieck, S.K., Goetze, B., Cui, L., Wong, M.H., Wenzel, C., Hosang, L., Dong, Y., Löwel, S., Schlüter, O.M. Progressive maturation of silent synapses governs the duration of a critical period. *Proc Natl Acad Sci U S A.* **112**, E3131-3140 (2015).

Josselyn, S.A., Frankland, P.W. Memory allocation: mechanisms and function. *Annu. Rev. Neurosci.* **41**, 389–413 (2018).

Lee, M.C., Yasuda, R., Ehlers, M.D. Metaplasticity at single glutamatergic synapses. *Neuron* **66**: 859–870 (2010).

Mammen, A.L., Kameyama, K., Roche, K.W., Huganir, R.L. Phosphorylation of the alpha-amino-3-hydroxy-5-methylisoxazole4-propionic acid receptor GluR1 subunit by calcium/calmodulin-dependent kinase II. *J. Biol. Chem.* **272**, 32528-32533 (1997).

Marin-Burgin, A., et al. Unique processing during a period of high excitation/inhibition balance in adult-born neurons. *Science* **335** (6073), 1238–1242 (2012).

Morrison, D.J., Rashid, A.J., Yiu, A.P., Yan, C., Frankland, P.W., Josselyn, S.A. Parvalbumin interneurons constrain the size of the lateral amygdala engram. *Neurobiol. Learn. Mem.* **135**, 91-99 (2016).

Nabel, E.M., Morishita, H. Regulating critical period plasticity: insight from the visual system to fear circuitry for therapeutic interventions. *Front. Psychiatry* **4**, 146 (2013).

Nicoll, R.A., Tomita, S., Brecht, D.S. Auxiliary subunits assist AMPA-type glutamate receptors. *Science* **311**, 1253-1256 (2006).

Roche, K.W., O'Brien, R.J., Mammen, A.L., Bernhardt, J., Huganir, R.L. Characterization of multiple phosphorylation sites on the AMPA receptor GluR1 subunit. *Neuron* **16**, 1179-1188 (1996).

Roth, B.L. DREADDs for Neuroscientists. *Neuron* **89**, 683-694 (2016).

Schnell, E., Sizemore, M., Karimzadegan, S., Chen, L., Brecht, D.S., Nicoll, R.A. Direct interactions between PSD-95 and stargazin control synaptic AMPA receptor number. *Proc Natl Acad Sci U S A.* **21**, 13902-13907 (2002).

Travaglia, A., Bisaz, R., Cruz, E. & Alberini, C. M. Developmental changes in plasticity, synaptic, glia and connectivity protein levels in rat dorsal hippocampus. *Neurobiol. Learn. Mem.* **135**, 125-138 (2016a).

Travaglia, A., Bisaz, R., Sweet, E.S., Blitzer, R.D. & Alberini, C.M. Infantile amnesia reflects a developmental critical period for hippocampal learning. *Nat. Neurosci.* **19**, 1225–1233 (2016b).

Travaglia, A., Steinmetz, A.B., Miranda, J.M. & Alberini, C.M. Mechanisms of critical period in the hippocampus underlie object location learning and memory in infant rats. *Learn. Mem.* **25**, 176-182 (2018).

REVIEWERS' COMMENTS:

Reviewer #1 (Remarks to the Author):

The authors have addressed my criticisms. I congratulate them for this outstanding paper.

Reviewer #2 (Remarks to the Author):

The authors have answered all my comments and have gone above and beyond to update the manuscript with novel data and discussions. I believe the paper is now ready for publication.

Reviewer #3 (Remarks to the Author):

The authors have addressed all my comments/suggestions in detail, and I believe that the manuscript is acceptable in the present form.

REVIEWERS' COMMENTS:

Reviewer #1 (Remarks to the Author):

The authors have addressed my criticisms. I congratulate them for this outstanding paper.

Reviewer #2 (Remarks to the Author):

The authors have answered all my comments and have gone above and beyond to update the manuscript with novel data and discussions. I believe the paper is now ready for publication.

Reviewer #3 (Remarks to the Author):

The authors have addressed all my comments/suggestions in detail, and I believe that the manuscript is acceptable in the present form.

Response to all Reviewers:

We thank very much the Reviewers for their helpful feedback, comments, and suggestions, which helped us to clarify and strengthen our manuscript.